# Single-atom level determination of 3-dimensional surface atomic structure via neural network-assisted atomic electron tomography

Juhyeok Lee [1], Chaehwa Jeong [1] & Yongsoo Yang [1 ✉]

Functional properties of nanomaterials strongly depend on their surface atomic structures, but they often become largely different from their bulk structures, exhibiting surface reconstructions and relaxations. However, most of the surface characterization methods are either limited to 2D measurements or not reaching to true 3D atomic-scale resolution, and single-atom level determination of the 3D surface atomic structure for general 3D nanomaterials still remains elusive. Here we demonstrate the measurement of 3D atomic structure at 15 pm precision using a Pt nanoparticle as a model system. Aided by a deep learning-based missing data retrieval combined with atomic electron tomography, the surface atomic structure was reliably measured. We found that <100> and <111> facets contribute differently to the surface strain, resulting in anisotropic strain distribution as well as compressive support boundary effect. The capability of single-atom level surface characterization will not only deepen our understanding of the functional properties of nanomaterials but also open a new door for fine tailoring of their performance.

[1] Department of Physics, Korea Advanced Institute of Science and Technology (KAIST), Daejeon, Korea. ✉email: yongsoo.yang@kaist.ac.kr

Precise determination of 3D surface atomic structure at an individual atom level has been the main interest for broad scientific communities including physics, materials science, chemistry, and nanoscience. Due to lower coordination numbers, surface atoms often show substantial deviations from their bulk structure[1–12]. However, especially for metallic nanoparticles, the surface structure plays a crucial role in their catalytic activities, which nowadays have major technological importance in the synthesis of chemicals[13], abatement of air pollution[14], and fuel cell applications[15]. It is critically important to fully understand the surface atomic structure to fine-tune the catalytic properties for each application.

Atomic electron tomography (AET) has been recently developed as a powerful tool for individual atom level 3D structural imaging[16,17], actively being used for measuring atomic level defects[18,19], 3D strain[19,20], chemical order/disorder[18], and nucleation dynamics[21]. However, often due to geometrical limitations, only part of a full tomographic angular range is experimentally measurable (so-called "missing wedge" problem), which results in elongation and Fourier ringing artifacts along the direction of the missing information in the reconstructed tomogram[22,23]. The missing wedge artifact negatively affects the accuracy of the surface atomic structure determined from the tomogram, being the main roadblock for precise determination of the 3D surface atomic structure[18].

On a parallel front, the deep learning-based neural network approach has recently attracted great interest from electron microscopists[24–27]. It has already demonstrated successes in missing data retrieval[25,26] and super-resolution imaging[24,27,28]. In this work, we combine AET with a deep learning-based neural network based on the atomicity principle. Using a Pt nanoparticle as a model system, we successfully retrieved the missing wedge information and achieved a robust reconstruction of the 3D surface atomic structure.

## Results

**Deep learning approach.** The main idea behind this deep learning-based approach is atomicity—the fact that all matter is composed of atoms. This means that the true atomic resolution electron tomogram should only contain sharp 3D atomic potentials convolved with the electron beam profile. Therefore, a deep neural network can be trained using simulated tomograms that suffer from artifacts (due to missing wedge, insufficient projection data, various noises, etc) as inputs, and the ground truth 3D atomic volumes as targets. The trained deep learning

network effectively augments the imperfect tomograms and removes the artifacts. Figure 1 shows our deep learning augmentation (DL augmentation) architecture based on a 3D-unet[29] (see "Methods" section). An input training data set was generated by simulating the electron tomography process using face-centered cubic (f.c.c.)-based random atomic models, and a target data set was prepared as the 3D volumes which consist of 3D Gaussian functions located at the ground truth atomic positions (see "Methods" section). For the tomography simulations, tomographic tilt series were obtained by linearly projecting the atomic potentials based on atomic scattering factors. The broadenings due to electron beam profiles and thermal vibrations were also considered during the tilt-series calculation by introducing a Bfactor (see "Methods" section). Only limited tilts angles (−65° to +65°) were used, and Poisson noises were added to simulate the experimental conditions. Three-dimensional tomograms were reconstructed from the tilt series using GENFIRE algorithm[30] (see "Methods" section). As shown in Fig. 1, the simulated tomograms suffer from undesirable artifacts along the vertical direction (missing wedge direction), and determination of true atomic structure from the raw tomogram is difficult due to these artifacts.

**Simulational test.** Our trained DL network was first tested with linear projection-based simulations. Using the atom-tracing method, the 3D coordinates of individual atoms were determined from the simulated tomograms (see "Methods" section). Compared to the ground truth structure (Fig. 2a, f, k), the raw tomogram clearly suffers from artifacts resulting from the missing wedge and noise effect; the atomic intensities are blurred, elongated, and connected to neighboring atoms, and several mis-identified atoms can be found, especially near the surface (Fig. 2b, g, l). Applying the atomicity-based DL augmentation can successfully suppress the artifacts (Fig. 2c, h, m). The atomic intensities become well-localized, and most of the atoms can be correctly retrieved. It can also be clearly seen from the Fourier peaks that the missing wedge information was successfully restored (Supplementary Fig. 1) by the DL approach.

To mimic the true experimental conditions including dynamic scattering, channeling, and lens aberrations, multislice-based plane-wave reciprocal-space interpolated scattering matrix (PRISM) simulations[31] were also performed to further test the DL augmentations (see "Methods" section). As expected, the raw tomograms from the PRISM simulation show more artifacts near the surface compared to the linear counterpart (Fig. 2d, i, n).

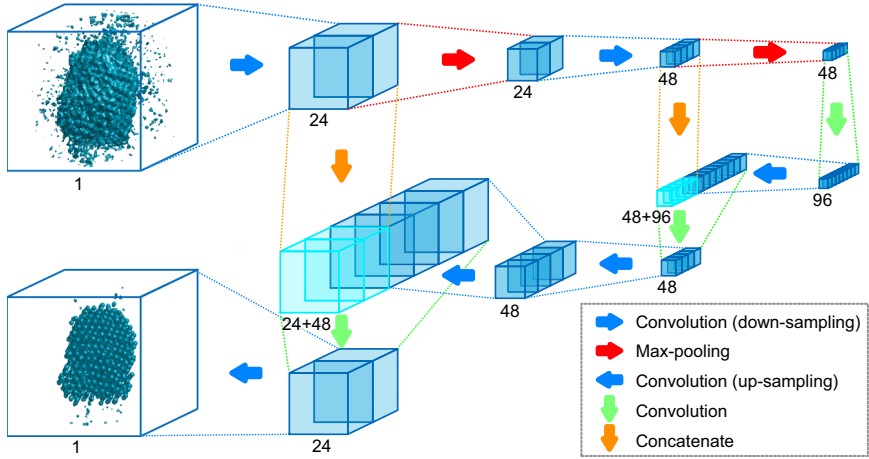

**Fig. 1 The architecture of the deep learning augmentation.** The deep learning augmentation follows a 3D-unet structure (see "Methods" section). The set of boxes represents the feature map. The number of channels is denoted below each feature map.

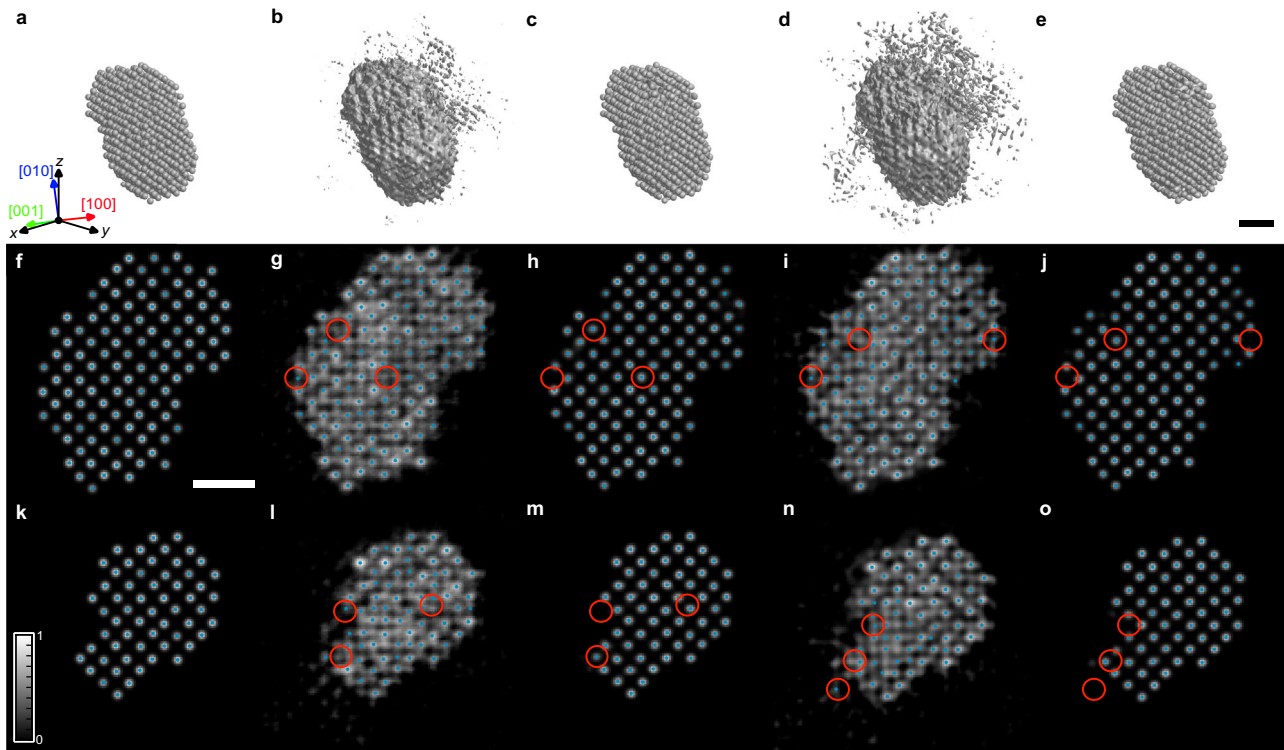

**Fig. 2 Effect of the DL augmentation for simulated tomograms. a–e** 3D iso-surfaces plotted with 10% iso-surface values (10% of the highest intensity), representing ground truth (**a**), linear tomogram before (**b**) and after the DL augmentation (**c**), PRISM tomogram before (**d**), and after the DL augmentation (**e**). Note that the z direction is the missing wedge direction. **f–o** 2-Å-thick slices perpendicular to [001] direction, obtained from the 3D tomograms near the center region (**f–j**) and near the surface (**k–o**). Ground truth (**f, k**), linear tomogram before (**g, l**) and after the DL augmentation (**h, m**), PRISM tomogram before (**i, n**) and after the DL augmentation (**j, o**). The grayscale background represents the reconstructed intensity, and blue dots represent the positions of traced atoms. Red circles denote misidentified atoms before the DL augmentation, which become correctly traced after the DL augmentation. Scale bars, 1 nm.

However, the DL augmentation was still successful in reducing the artifacts even for PRISM-simulated tomograms, and the output volumes were as similar as the ones obtained from the linear simulations (Fig. 2e, j, o).

Since the traced atomic coordinates can be quantitatively compared to the ground truth atomic model, we performed a statistical analysis based on two test data sets of 1000 tomograms, each generated from linear simulations and PRISM simulations, respectively. The averaged tracing errors (percentage of incorrectly identified atoms; see "Methods" section) of 6.8% (linear) and 7.9% (PRISM) of the raw tomograms were reduced to 0.4% and 0.7% after the DL augmentation, respectively (Supplementary Fig. 2a, b). Also, the before-DL augmentation averaged root-mean-square deviations (RMSDs) between the ground truth atomic positions and traced atomic positions were 34.5 pm (linear) and 36.5 pm (PRISM), which were substantially improved to 19.7 and 22.3 pm after the DL augmentation, respectively (Supplementary Fig. 2e, f).

To verify the robustness of our approach, the network was trained with two additional training data sets (one with amorphous structure and one with f.c.c.-based structure with a different Bfactor [see "Methods" section]), and tested with the same test data set used for testing the original DL augmentation. As can be seen in Supplementary Fig. 2, regardless of the base structure of the training data sets, all DL augmentations show consistent output, and proper f.c.c.-based ground truth structures of the test data set were retrieved even by the DL augmentation trained with amorphous structures. Further, we applied the DL augmentation to simulated tomograms from various atomic models to check whether our approach is also

valid for structures other than close-to-ideal f.c.c. We generated the test data sets based on the following atomic structures: f.c.c.-based atomic structures with the different vacancy defect levels, amorphous atomic structures with different sizes, decahedral nanoparticles with twinned boundaries, and f.c.c.-based atomic structures with stronger surface relaxations (see "Methods" section). The DL augmentation network shows significant improvement regardless of the base-model of the input volumes (Supplementary Figs. 3–9), supporting the robustness of the DL augmentation approach.

To evaluate the performance of our DL augmentation specifically in terms of surface structure, we calculated the tracing errors and RMSDs for the surface atoms before and after the DL augmentation. For the linear projection simulation case, the averaged surface tracing error in Supplementary Fig. 2c decreased from 4.4 to 0.2%, and the averaged surface RMSD was reduced from 30.7 to 18.0 pm (Supplementary Fig. 2g). PRISM simulations also showed substantial improvement (the averaged surface tracing error 0.6%, and averaged surface RMSD 21.1 pm after the DL augmentation). These simulation results clearly demonstrate that the DL augmentation can successfully reduce the artifacts from insufficient data and noises, retaining improved precision, especially for surface atoms. The DL augmentations trained with different training data sets also showed similar improvements (Supplementary Fig. 10).

**Experimental determination of the 3D atomic structure of a Pt nanoparticle.** Next, we applied our DL augmentation to experimentally determine the 3D surface atomic structure of a Pt

nanoparticle. The experiment was performed using an aberration-corrected scanning transmission electron microscope (STEM) operated in annular dark-field (ADF) mode (see "Methods" section). From a 4 nm diameter Pt nanoparticle, a tilt series of 21 images were acquired with the tilt angles ranging from −71.6° to +71.6° (Supplementary Fig. 11). After image post-processing, a 3D tomogram was reconstructed from the tilt series using GENFIRE algorithm[30] (see "Methods" section). Figure 3a shows the raw 3D reconstruction of the nanoparticle. Atom-tracing and classification procedures were applied to the volume, resulting in a 3D atomic model of 1411 Pt atoms (Fig. 3c and see "Methods" section).

Severe artifacts due to the missing wedge problem and noises can be clearly seen in Fig. 3a and Supplementary Figs. 12a, e, i, 13a–d. The reconstruction suffers from elongation and undesired intensity reduction near the surface, especially along the missing wedge direction. Although many of the atoms are correctly found at f.c.c. lattice sites, some of the atoms are clearly misidentified and surface atoms are not well defined (Fig. 3a, c and Supplementary Fig. 13). A 3D mask is often employed to define the surface of nanoparticles in this case. However, the mask depends on threshold parameters. Supplementary Fig. 13 shows the parameter dependence of the mask.

To identify the precise atomic structure including the surface, the DL augmentation was applied to the raw tomogram. Figure 3b, d clearly shows that the atomic intensities in the DL-augmented output are well isolated with the expected Gaussian shape, showing drastic improvement compared to the raw reconstruction. Atom-tracing on the DL-augmented volume resulted in 1530 atoms; about 100 more atoms were successfully identified (Fig. 3d). Several missing atoms near the core region were restored by the DL augmentation (Fig. 3d and Supplementary Fig. 14). Atom profiles (Supplementary Figs. 15, 16) clearly show that the elongation along the missing wedge direction is successfully resolved by the DL augmentation. The expected f.c.c. Fourier peak structures were also well-retrieved after the DL augmentation especially along the missing wedge direction (Supplementary Fig. 12).

We further calculated the forward projection tilt-series images from the atomic models obtained from the raw and DL-augmented tomograms and compared them to the experimental tilt series by calculating the R-factor[18,19,21] (see "Methods" section). The R-factor was improved from 0.192 to 0.174 after the DL augmentation. Furthermore, the surface boundaries, especially along the missing wedge direction, are now clearly defined after the DL augmentation, which allows unambiguous determination of 3D surface atomic structure without parameter-dependent masking process (Supplementary Fig. 13).

PRISM tomography simulation of the atomic model obtained from the DL-augmented volume demonstrated 98.8% accuracy of atom identification and 15.1 pm precision of the atomic coordinates (see "Methods" section). We quantitatively analyzed the improvement made from the DL augmentation (Supplementary Fig. 17). The atomic structures before and after the DL augmentation showed a difference of 373 atoms (25.4%). About 60% of the difference comes from the surface (251 atoms). This mainly results from the unidentifiable surface atoms (due to missing wedge-induced intensity drop) being successfully traced after the DL augmentation (Supplementary Fig. 14). We further verified our approach by applying the two other DL augmentations trained by the amorphous atomic models and the f.c.c. atomic models with sharper Gaussian widths (i.e., smaller Bfactor) (see "Methods" section). The differences between the atomic models obtained from three different DL augmentations showed a higher consistency compared to the differences between the models before and after the DL augmentation (Supplementary

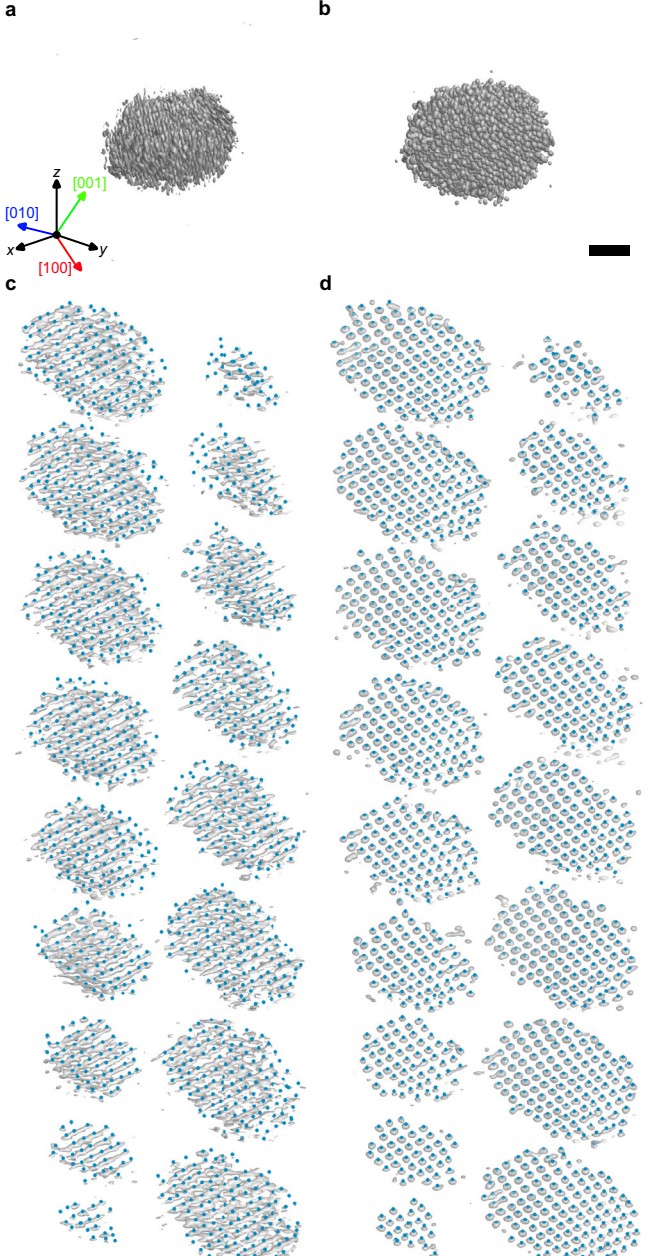

**Fig. 3 The 3D density maps of the experimentally measured Pt nanoparticle tomograms and traced atomic coordinates. a, b** Iso-surfaces of reconstructed 3D density are plotted with 40% and 15% iso-surface values from the maximum intensity for tomograms before (**a**) and after (**b**) the DL augmentation, respectively. The z direction is the missing wedge direction. **c, d** Reconstruction volume intensity and traced atom positions. Each slice represents an atomic layer, and the blue dots indicate the traced 3D atomic positions before (**c**) and after (**d**) the DL augmentation. The grayscale backgrounds of the atomic positions are iso-surfaces of 3D density with 40% (**c**) and 15% (**d**) iso-surface values from the maximum intensity. The sliced layers are perpendicular to [001] direction. Here the DL augmentation network trained by the f.c.c.-based atomic model with Bfactor 5 Å² was used. Scale bars, 1 nm.

Fig. 17). Also, the missing Fourier peak structures expected for f.c.c. were properly reconstructed even with the DL augmentation trained by the amorphous atomic models (Supplementary Fig. 12c).

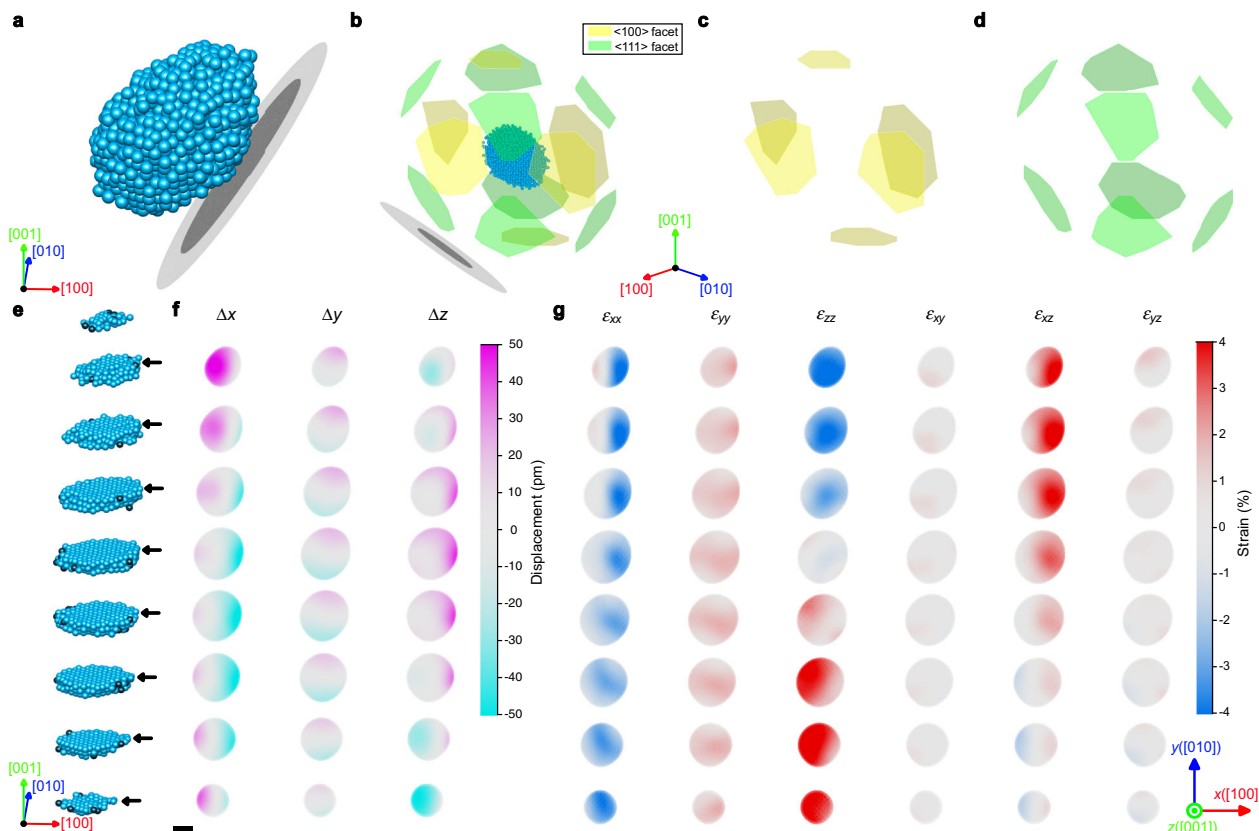

**Fig. 4 Facets, 3D atomic displacements, and strain maps of the Pt nanoparticle. a** Overall atomic structure of the Pt nanoparticle with SiN substrate represented as black and gray disks. **b–d** Identified facet structure of the Pt nanoparticle, showing all facets (**b**), <100> facets (**c**), and <111> facets (**d**). **e** Atomic structure of the Pt nanoparticle represented in **a** divided into layers of one f.c.c. unit cell thickness. Blue and black atoms represent the atoms assigned and not-assigned to the ideal f.c.c. lattice sites, respectively. **f** The atomic displacements along the crystallographic axes. **g** The strain maps representing six components ($\varepsilon_{xx}$, $\varepsilon_{yy}$, $\varepsilon_{zz}$, $\varepsilon_{xy}$, $\varepsilon_{xz}$, $\varepsilon_{yz}$) of the strain tensor. The atomic displacements and strain tensors in each row were calculated from the corresponding slice pointed by the black arrows in **e**. These calculations were based on the traced atomic model from the tomogram after applying the DL augmentation trained by f.c.c.-based atomic models with Bfactor 5 Å$^2$. Scale bar, 2 nm.

**Strain analysis**. Having accurate 3D atomic coordinates directly yields the 3D displacements and strain tensor. By comparing with an ideal f.c.c. lattice, the 3D displacements, and strain tensor were calculated based on the traced atomic models (see "Methods" section). The out-of-plane atomic displacements in <100> and <111> facets (Fig. 4a–d, f) were −10.2 ± 62.9 pm and −3.3 ± 41.2 pm, respectively. However, part of the surface of the nanoparticle was making contact with the SiN membrane substrate (Fig. 4a). To understand the substrate effect, displacements of the atoms on the facets making contact with the substrates were separately calculated, resulting in the out-of-plane displacements of −17.2 ± 86.5 pm for <100> facets and −21.7 ± 47.4 pm for <111> facets. For facets not in contact with the substrates, the average out-of-plane displacements of −6.4 ± 45.6 pm (compressive) and 5.3 ± 35.0 pm (tensile) were obtained for <100> and <111> facets, respectively. This behavior (compressive strain for <100> facets and tensile strain for <111> facets) is consistent with the theoretical calculation result[3].

The strain map (Fig. 4g) shows strong compressive strain along the *x* direction and tensile strain along the *y* direction. Interestingly, the strain along the *z* direction shows both compressive (near the [001] facet) and tensile strain (near the [00$\bar{1}$] facet). The anisotropic strain behavior is likely to be related to the shape of the nanoparticle as well as the particle–substrate interface. Therefore, we conducted a shape analysis by assuming an ellipsoidal shape of the nanoparticle (see "Methods" section). The vertical direction (the shortest ellipsoid principal axis) of the

nanoparticle is slightly tilted compared to the lab-coordinate *z* direction (i.e., electron beam direction), as indicated in Supplementary Fig. 18. When the nanoparticle is re-oriented based on the f.c.c. crystallographic axes for the strain calculation, the contact between the nanoparticle and the SiN substrate becomes located at the lower right part of the nanoparticle (Fig. 4a).

To clarify the exotic strain behavior, we plotted the strain map in the lab-coordinate (Supplementary Fig. 19g). Tensile strain is dominant along the *x* and *y* directions in the lab-coordinate, and strong compressive strain only occurs near the interface with the SiN membrane (along the *z* direction in the lab-coordinate). This indicates that the strong compressive strain observed along with the [100] direction (Fig. 4g) is mainly due to the particle–substrate interface effect. We also found that the tensile strain observed along [0 ± 10] directions is resulting from the neighboring <111> facets which show clear tensile strain ($\varepsilon_{xx}$ maps in Supplementary Fig. 20). This explains the opposite strain behavior along the *x* and *y* directions in Fig. 4g. Also, the strain map in the lab-coordinate (Supplementary Fig. 19g) shows that the *z* directional strain $\varepsilon_{zz}$ exhibits a gradual change from tensile (the lower right part of the nanoparticle) to compressive (the upper left part). Although compressive strain is expected along *z* direction due to the domination of <100> facets, strong tensile strain is observed at the particle–substrate interface. This result indicates that the nanoparticle surface structure is strongly dependent on the boundary condition, and the choice of support

material can critically influence the structure, strain, and related catalytic behavior.

The strain maps were also calculated from the atomic structures obtained from the raw tomogram (not DL-augmented) and tomograms augmented with two other differently trained DL networks (using amorphous models and smaller Bfactor f.c.c. models). Some differences can be found (especially near the surface) between the raw case and DL-augmented cases, but the strain maps from DL-augmented tomograms are very consistent overall, verifying the robustness of the DL augmentation approach (Supplementary Fig. 19).

## Discussion

In summary, the 3D atomic structure of a nanoparticle was successfully determined at an individual atom level by neural network-assisted AET. Using a Pt nanoparticle as a model system, we demonstrated that the atomicity-based approach can reliably identify the surface atomic structure with a precision of 15 pm. The atomic displacement, strain, and facet analysis revealed that the surface atomic structure and strain are related to not only the shape of the nanoparticle but also the particle–substrate interface. Combined with quantum mechanical calculations such as density functional theory, the capability of precise identification of surface atomic structure will serve as a powerful key to understand the surface/interface properties such as catalytic performance and oxidation effect.

## Methods

**Generation of input and target data sets for neural network training.** To train and test the DL neural network, 12,000 (10,000 for training, 1000 for validation, and 1000 for test) simulated tomograms (for input data) and ground truth 3D tomograms (for target data) were generated. Simulated tomograms were prepared by following a four-step process.

(1) Random-shaped 3D volumes without cavity were created, whose volumes range from 17,000 to 27,000 $Å^3$. Then, atoms were placed within the 3D volumes based on f.c.c structure with lattice constant 3.912 Å in random orientation. From the atomic models, to mimic atomic structures of realistic nanoparticles, some percentages of atoms (the percentages randomly chosen between 0 and 0.5%) were randomly removed to simulate atomic defects, and random spatial displacement of ~22 pm RMSD was also applied to each atom.

(2) 3D volumes of atomic potentials were calculated based on the 3D atomic structures. The atomic potentials were obtained by Fourier transformation of the electron scattering factors[32]. The atomic potentials were then convolved with Gaussian kernels to consider the thermal vibration and electron beam size effect. The standard deviation ($\sigma$) of a Gaussian kernel is $(Bfactor/2\pi^2)^{0.5}$. The Bfactors were randomly selected from a Gaussian distribution with a mean 5 $Å^2$ and standard deviation of 1 $Å^2$. Mean Bfactor of 5 $Å^2$ showed the best consistency between the simulated projections and experimental tilt-series images.

(3) Tilt series were generated by taking linear projections from the 3D atomic potential volumes at each tilt angle. Each tilt series was composed of 21 projections with tilt angles ranging from −65° to +65°. The size of each projection was 144× 144 pixels with a pixel size of 0.357 Å. Poisson noises were added to the projections to simulate the noise effect in the experiment. To take into account the angular error due to stage instability during the experiment, random angle errors up to ±0.3° were also added to each tilt angle.

(4) 3D tomograms were reconstructed from the tilt series and corresponding tilt angles using GENFIRE algorithm[30]. The axis convention was chosen so that the z direction is the missing wedge direction. For GENFIRE reconstructions, the fast Fourier transform (FFT) interpolation method, number of iterations 100, oversampling ratio of 2, and interpolation radius 0.3 were used. The target data set (ground truth) was generated by following step 1 above. At the position of each atom, a Gaussian intensity distribution with a standard deviation ($\sigma$) that corresponds to Bfactor 4 $Å^2$ was placed, which showed the best performance in our simulation tests.

To verify the robustness of the DL augmentation approach, two other training data sets were generated and tested: one with f.c.c.-based structures with Bfactor 3.2 $Å^2$, and another based on amorphous atomic structures instead of f.c.c., following the same four-step process described above. For the amorphous structure case, the volumes ranged from 23,000 to 34,000 $Å^3$, and the atomic positions were randomly placed within the volume with the constraint of minimum distance 2.0 Å, until 99% atom density compared to that of the f.c.c. structure was reached.

**Generation of additional test data sets for testing the DL network.** We generated additional test data sets based on four-different atomic models: f.c.c.-based

atomic models with different levels of vacancy defects, small (approximately 1 nm) amorphous atomic models, atomic models of decahedral shaped nanoparticles with twinned boundaries, and f.c.c.-based atomic models with different spatial displacements for the core and the surface atoms.

In the case of the f.c.c.-based atomic models with different levels of vacancy defects, we simulated three different test data sets using vacancy defect levels of 5%, 10%, and 20%, respectively. We followed the same procedure of generating the data set of f.c.c.-based atomic models with Bfactor 5 $Å^2$; the only difference here is the level of vacancy defects, which was randomly chosen between 0% and 5%, 10%, and 20%, respectively. Second, the small amorphous models (about 1 nm diameter) were generated following the procedure of generating the amorphous atomic model with Bfactor 5 $Å^2$. The only difference is that each volume ranges from 3400 to 5700 $Å^3$ (about 200–300 atoms) in this case. Third, an atomic model of decahedral-shaped Pt nanoparticle with twinned boundaries[33] was used for generating different tomograms by applying random cropping/rotation/translation and 0.5% vacancy defect insertion. Fourth, to mimic the true surface structure of Pt nanoparticles, we generated f.c.c.-based atomic models with stronger spatial displacements near the surface. Following the same procedure of generating the f.c.c.-based atomic models with Bfactor 5 $Å^2$, the spatial displacements of 22 and 50 pm RMSD were randomly applied to the core and surface atoms, respectively. We generated the data sets of 100 tomograms for each model. Note that for the simulation of the decahedral-shaped Pt nanoparticle, 27 tilt-series images were used (instead of 21) and the tilt angles were chosen in the range between −75° and +75° because resolving twin boundary structures required a higher-quality tomogram. These atomic models were used as test data sets to check whether the DL augmentation network is biased toward the base atomic structure of the training data sets (Supplementary Figs. 3–8), and the result clearly shows that the trained DL augmentation network works well even for test data sets with atomic structures different from those of the training data sets.

**Bfactor optimization for the training input data.** The width of Gaussian broadening resulting from the electron beam size and thermal vibration was determined to generate training data consistent with the experimental data. A 3D Gaussian function of 11× 11× 11 voxels was fitted to the averaged 11× 11× 11 voxels extracted from the local maxima positions of the experimental 3D tomogram of a Pt nanoparticle. The optimized Bfactor ($\sigma^2 \times 2\pi^2$) was determined to be 5 $Å^2$.

**PRISM STEM simulation for the test data set.** Following the same convention with the training data generation, 1000 simulated tilt series were generated by PRISM simulation[31,34]. The PRISM calculations were done with the following parameters: 300 keV electron energy, −775 nm $C_3$ aberration, 378 μm $C_5$ aberration, 25.1 mrad convergence semi-angle, 40 and 200 mrad detector inner and outer semi-angle, interpolation factor 8 and 2 Å slice thickness. Each tilt image was convolved with a Gaussian function with Bfactor 5 $Å^2$. Also, random angle errors up to ±0.3° were added to each tilt angle. 3D reconstructions were calculated from the obtained tilt series using GENFIRE algorithm[30] with the same reconstruction parameters given above.

**Deep learning augmentation architecture and training.** The sizes of input and output data for the neural network-based DL augmentation were both 144× 144× 144 voxels. Figure 1 shows our neural network structure based on a 3D-unet. For computational efficiency, the network was designed to reduce the size of feature maps as the layer gets deeper. The encoder consists of two 3× 3× 3 convolutions with stride 2 and two 2× 2× 2 max-poolings for downsampling. The bridge linking the encoder and decoder consists of one 3× 3× 3 convolution with stride 1. The decoder consists of four 3× 3× 3 transposed convolutions with stride 2 for upsampling and two 3× 3× 3 convolutions with stride 1. Leaky Rectified Linear Unit (LReLU)[35] with the coefficient of leakage 0.2 was used as an activation function except for the connections from the last layer toward the final output, where regular ReLU[36] was used. Dropout[37] method was used for the 1st, 5th, and 9th layers to prevent overfitting. Loss function was the mean-square error, and Adam optimizer[38] was used with the learning rate of $2 \times 10^{-4}$.

Three different neural networks (i.e., DL augmentations) were trained by three different input and target data sets, respectively, which are the data sets generated by (i) f.c.c.-based atomic model + Bfactor 5 $Å^2$, (ii) amorphous-based atomic model + Bfactor 5 $Å^2$, and (iii) f.c.c.-based atomic model + Bfactor 3.2 $Å^2$.

To choose a proper number of tomographic 3D volumes for the training, we tested various input volumes and found that the use of more than 5000 volumes is enough to obtain consistent output. Supplementary Figure 21 shows the evolution of validation losses during the training based on 5000 and 10,000 training volumes, and they show a similar level of the loss function. We, therefore, used 10,000 test volumes for the training of the DL augmentation network.

To prevent overfitting and find a proper number of epochs for the training, we monitored the learning curves during the training processes. The total number of epochs was set to 100 for each training. The learning curves in Supplementary Fig. 22 show that the training processes of all three DL networks were successfully converged after ~50–70 epochs with no signs of overfitting.

**STEM data acquisition**. Bare Pt nanoparticles were purchased from nano-Composix, supplied in aqueous 2 mM sodium citrate. The solution was drop cast onto a 5 nm thick SiN membrane grid and annealed in vacuum at 150 °C for 24 h. The tomographic tilt series were acquired using a Titan Double Cs corrected transmission electron microscope (Titan cubed G2 60-300). The images were collected at 300 kV in annular dark-field scanning transmission electron microscopy mode with 40 mrad and 200 mrad detector inner and outer semi-angles, respectively. The convergence beam semi-angle was 25.1 mrad. Total 21 tilt-series images for tilt angles ranging from −71.6° to 71.6° were acquired. For each tilt angle, three consecutive 1024 × 1024 images were measured with 4 $\mu$s dwell time with 15.1 pA beam current, and the pixel size was 0.357 Å. The total electron dose for the entire tilt series was $1.86 \times 10^5$ eÅ$^{-2}$. To check the possibility of structural change due to the electron beam during the tilt-series acquisition, the zero-degree projection was measured again right after the experiment. The particle was rotated a little bit (Euler angles ($z$–$y'$–$x''$ convention) $\{\psi : 4°, \theta : -3.5°, \varphi : 1°\}$) during the experiment, but the post-experiment zero-degree projection was consistent with the linear projection of the final atomic model obtained from AET (after applying the rotation), evidencing that the atomic structure did not change during the experiment (Supplementary Fig. 23).

**Image post-processing, GENFIRE reconstruction, tomogram post-processing**. Image post-processing (drift correction, scan distortion correction, BM3D noise reduction[39,40], tilt-series alignment based on center-of-mass and common line method) was conducted as described in previous works[18–21]. Also, to remove undesirable high-frequency noise, a circle-shaped low-pass filter with a diameter of 2.35 Å$^{-1}$ (the edge of the filter was slightly smoothened by 0.07 Å$^{-1}$) was applied to the images before the BM3D noise reduction process.

After the image post-processing, a 3D reconstruction was calculated using GENFIRE algorithm[30]. To reduce tilt angle error and improve reconstruction quality, GENFIRE-based angular refinement and spatial re-alignment were also conducted[21,30]. After angular refinement, the final 3D reconstruction was obtained by running GENFIRE algorithm[30] with the following parameters: discrete Fourier transform (DFT) interpolation method, number of iterations 1000, oversampling 4, and interpolation radius 0.1. To remove salt-pepper-like high-frequency noise, upsampling and downsampling (binning) were subsequently applied. To match the intensity scale between the experimental tomogram and the tomograms used for the neural network training, a scale factor was calculated. Integrated intensities for $3 \times 3 \times 3$ voxels near all traced atom positions were averaged for both experimental and simulated 3D volumes, respectively. The ratio of the two averaged integrated intensities was used as the scale factor and applied to the experimental tomogram before running the DL augmentation. A DL-augmented tomogram was obtained by applying the DL augmentation trained by the f.c.c.-based data set with Bfactor 5 Å$^2$.

**Identification of 3D atomic coordinates: atom-tracing**. From the raw and DL-augmented 3D tomograms, 3D atomic coordinates were obtained by the atom-tracing procedure[18,21] as follows. First, all local maxima positions were found from the 3D tomogram, and sorted by the intensity in descending order from the highest intensity one. Second, after cropping $5 \times 5 \times 5$ voxels centered on each local maximum, a 3D Gaussian function was fitted for each cropped volume. Starting from the highest intensity local maximum, the fitted position was added to the traced atom list if all the distances between the position of the newly fitted maximum and the positions already in the list were larger than 1.8 Å.

Obtained local maxima were the mixture of proper Pt atoms and weak intensity spots (non-atoms) originating from reconstruction artifacts. The local maxima were classified into proper Pt atoms and non-atoms by following the unbiased classification method described in refs. [18,21].

To finalize the 3D atomic structure from the DL-augmented tomogram, a manual correction was applied to add (or remove) physically (un)reasonable atom candidates. A minimum distance constraint of 1.6 Å was used during this process. Total 8 atoms were manually added, and 9 atoms were manually removed. For sanity check, the same atom-tracing procedure was applied to the tomograms augmented with two differently trained networks (trained with the amorphous model-based data set, trained with the f.c.c.-based data set with Bfactor 3.2 Å$^2$). For these results, 7 and 8 atoms were manually added, and 32 and 5 atoms were manually removed, respectively.

To measure the error between the experimental and simulated STEM images, R-factor[18,19,21] was used. We calculated the R-factor between the experimental tilt series and linear projections of the 3D final atomic models from the tomogram after the DL augmentation trained by the f.c.c.-based atomic models with Bfactor 5 Å$^2$ (Supplementary Fig. 23). The averaged R-factor was 0.173. Additionally, averaged R-factors were calculated by comparing the experimental images and linear projection images of the atomic structures from the tomograms augmented by two different DL networks (trained by the amorphous-based atomic models with Bfactor 5 Å$^2$ and the f.c.c.-based atomic models with Bfactor 3.2 Å$^2$). They were determined to be 0.182 and 0.176, respectively. Note that the atomic structures obtained from the DL-augmented tomograms were manually corrected, as described above. To properly compare the results before and after the DL augmentation, we also calculated the R-factors from the atomic structures before and after the DL augmentation without the manual corrections. The averaged R-factors from the atomic structures from the raw tomogram and the tomograms

after applying three different DL networks (trained by the f.c.c.-based atomic models with Bfactor 5 Å$^2$, the amorphous-based atomic models with Bfactor 5 Å$^2$, and the f.c.c.-based atomic models with Bfactor 3.2 Å$^2$) were determined to be 0.192, 0.174, 0.182, and 0.177, respectively. The R-factor is clearly decreased after the DL augmentation, and also the R-factors obtained from the manually corrected atomic models are better than those from the uncorrected ones. The atomic structure obtained from the DL augmentation trained by the f.c.c.-based atomic structures with Bfactor 5 Å$^2$ showed the best consistency with the experimental STEM images.

**Defining the surface atoms**. For surface analysis, the surface atoms were determined using alpha shape algorithm[41] with shrink factor 0.5. This method was applied for both simulated and experimental data. For the experimentally measured Pt nanoparticle, the numbers of the surface atoms were 248 and 419 atoms before and after the DL augmentation, respectively.

**Calculation of tracing error**. We defined an error between two atomic coordinate sets (so-called tracing error). To match common atom pairs, a distance threshold of 1.2 Å (smaller than half of Pt covalent bonding length) was used. The number of uncommon atoms (and surface uncommon atoms) was defined as the total sum of the number of (surface) atoms in the first atomic model but not in the second atomic model plus the number of (surface) atoms in the second atomic model but not in the first atomic model. Then, the (surface) tracing errors were calculated as the ratio of the number of (surface) uncommon atoms to the number of total (surface) atoms in the ground truth atomic model (Supplementary Figs. 2, 3, 5, 7–9). If ground truth is not available (as in the case of experimental data), the ratio was calculated to the average of the total number of atoms of the two atomic models being compared (Supplementary Figs. 10 and 17).

**Precision analysis for experimental data by PRISM STEM simulation**. To estimate the precision of the atomic coordinates obtained from our approach, we ran a precision analysis[18–20] using PRISM simulation[31,34]. A tilt series of 21 projection images were calculated from the experimentally determined 3D atomic model by PRISM simulation[31,34] using the same parameters described above with interpolation factor 2, without adding angular errors. A 3D reconstruction was calculated using the GENFIRE algorithm[30]. A corrected tomogram was obtained by applying the DL augmentation to the 3D reconstruction. Atomic models were obtained from both the raw and DL-augmented tomograms by applying the atom-tracing procedure. The numbers of traced atoms from the PRISM tomograms before and after applying the DL augmentation were 1494 and 1511 atoms, respectively. The percentage of common atoms (accuracies of atom identification) between the original 3D atomic model and the PRISM traced models were 96.5% (1476 atoms) and 98.8% (1511 atoms), for before and after the DL augmentation, respectively. The RMSDs of common atoms pairs (the precisions of our atomic structures) were 26.1 (before DL) and 15.1 pm (after DL).

**Facet analysis, displacement field, and strain map calculation**. The atomic displacements and 6 components of the 3D strain tensor ($\varepsilon_{xx}$, $\varepsilon_{yy}$, $\varepsilon_{zz}$, $\varepsilon_{xy}$, $\varepsilon_{xz}$, $\varepsilon_{yz}$) were calculated by comparing the experimentally observed 3D atomic structure with an ideal f.c.c. lattice[20]. The 3D atomic structure was assigned to ideal f.c.c. lattice with the following procedure. First, a starting atom was chosen as an origin of an f.c.c. lattice. Then, ideal Pt f.c.c. nearest neighbor positions of the atom were calculated. For each nearest neighbor position, if there was an atom within the distance 0.76 Å (27.5% of Pt covalent bonding length), the atom was added to the corresponding f.c.c. lattice site. The nearest neighbor search was repeated for all the newly assigned f.c.c. lattice sites. The process was repeated until no more atoms could be assigned to the lattice. Second, a new f.c.c. lattice was fitted (fitting parameters: translation, 3D rotation and lattice constant) to the atoms assigned to the lattice. These two steps were iterated using the newly fitted f.c.c. lattice until the difference between the old and newly fitted lattice constant was less than $10^{-5}$ Å. After this process, 97.1% of the atoms were successfully assigned to f.c.c. lattice sites, and the RMSD between the assigned atom positions and the fitted f.c.c. lattice was 64.01 pm. The fitted f.c.c. lattice constant was 3.85 Å.

<100> and <111> facets were defined as the outermost atomic planes perpendicular to <100> or <111> directions, which include more than 10 atoms. The total number of atoms on <100> and <111> facets were 160 and 226, respectively. The out-of-plane and in-plane atomic displacements between the fitted f.c.c. lattice and assigned atom positions for atoms on each facet were calculated. Averaged out-of-plane atomic displacements of <100> and <111> facets were determined to be −10.2 ± 62.9 pm and −3.3 ± 41.2 pm, respectively (minus sign means inward displacement, and the uncertainty means the standard deviation of the atomic displacements). For averaged in-plane displacements, 55.4 ± 33.3 pm and 65.0 ± 34.6 pm were observed for <100> and <111> facets, respectively. Comparing the number of atoms in <100> and <111> facets, it was found that the numbers of atoms on [100], [$\bar{1}$00], [010], [0$\bar{1}$0], [001] and [00$\bar{1}$] facets were 35, 32, 34, 22, 16 and 21, respectively. And, the numbers of atoms on [111], [11$\bar{1}$], [1$\bar{1}$1], [$\bar{1}$11], [1$\bar{1}\bar{1}$], [$\bar{1}$1$\bar{1}$], [$\bar{1}\bar{1}$1] and [$\bar{1}\bar{1}\bar{1}$] facets were 27, 27, 20, 15, 45, 37, 25 and 30, respectively. We found that the facets close to the particle–substrate interface

([100], [00$\bar{1}$], [11$\bar{1}$] and [1$\bar{1}\bar{1}$] facets) are mainly responsible to the strong compressive strain (Fig. 4 and Supplementary Figs. 19, 20).

The atomic displacement field was mapped on a cubic grid by applying a Gaussian kernel with a standard deviation of 5.5 Å to the measured atomic displacements. 3D strain tensor maps were obtained by taking derivatives from the displacement map.

A similar analysis was performed on the atomic structures from the raw tomogram, tomogram augmented by the amorphous model-trained DL network, and tomogram augmented by the f.c.c. model with Bfactor 3.2 Å$^2$ model-trained DL network. The fraction of atoms successfully assigned to f.c.c. lattice sites were 91.3%, 95.9%, and 97.6%, respectively. The RMSDs between each fitted f.c.c. lattice and the given atomic structures were 86.35, 73.74, and 64.39 pm, respectively.

**Ellipsoid fitting for Pt nanoparticle**. For analysis of the shape of the Pt nanoparticle, we fitted an ellipsoid to the determined surface atomic structure. To find the closest ellipsoid, the positions of the defined surface atoms were exploited to fit the parameters in the ellipsoid equation[42]. The fitted principal semi-axes in the ellipsoid were determined to be [0.92, 0.36, 0.15], [0.38, −0.92, −0.11], and [−0.10, −0.15, 0.98] (in the lab-coordinates) with corresponding lengths of 20.0, 16.9, and 13.6 Å, respectively.

## Data availability

All of our experimental data, tomographic reconstructions, determined atomic structures are posted at http://mdail.kaist.ac.kr/DLaugmentation.

## Code availability

All neural network-related codes and data set generating source codes are available at http://mdail.kaist.ac.kr/DLaugmentation.

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

## Acknowledgements

We thank Chang Yun Son and Aloysius Soon for helpful discussions. This research was supported by the National Research Foundation of Korea (NRF) Grants funded by the Korean Government (MSIT) (Nos. 2019R1F1A1058236 and 2020R1C1C100623911). J.L. and C.J. were also partially supported by the KAIST-funded Global Singularity Research Program (M3I3) for 2019, 2020, and 2021. The STEM experiment was conducted using a double Cs corrected Titan cubed G2 60-300 (FEI) equipment at KAIST Analysis Center for Research Advancement (KARA). Excellent support by Hyung Bin Bae, Jin-Seok Choi, and the staff of KARA is gratefully acknowledged.

## Author contributions

Y.Y. conceived the idea and directed the study. J.L. designed the neural network and performed simulational data analysis. Y.Y. designed and performed the experiment. C.J. and J.L. conducted the experimental data analysis. J.L. and Y.Y. wrote the manuscript. All authors commented on the manuscript.

## Competing interests

J. L., C. J. and Y. Y. have patent applications (10-2020-0176633 [Korea] and 17/132,762 [US]), which disclose the DL augmentation method for 3D atomic level structural determination based on AET.
