## [Peer Review File · Nature Communications]

REVIEWER COMMENTS

Reviewer #1 (Remarks to the Author):

Report on: Single-atom level determination of 3-dimensional surface atomic structure via neural network-assisted atomic electron tomography
By Juhyeok Lee, Chaehwa Jeong and Yongsoo Yang

This article demonstrates a quite spectacular increase in the fidelity of three-dimensional reconstruction of fcc metallic nanoparticles using a machine learning approach in combination with relatively sparse aberration-corrected scanning transmission electron microscopy (STEM) tomograms. The improvement is primarily a result of the application of a three-dimensional unet machine-learning CNN algorithm, trained on PRISM-simulated data.

The reconstructed data is certainly impressive, but a great deal of information is lacking that would convince the reader that the results are reliable. I also find the description of the procedure used as a 'machine-learning filter' to be misleading and unsatisfactory. A filter, in the usual sense of the word, removes some component, separating it from another component that is of more use or interest. This applies to filtration of materials from liquids, removal of contaminants, improving signal to noise ratio or choosing a particular frequency, or energy, of some radiation or other signal. The process here is quite different; here, a machine learning algorithm fabricates some imagined data to fill gaps in a sparse data set. The fidelity of this process, when applied to test or experimental data, relies on the accumulated statistical accuracy of performing the same process on a large training data set. The process is thus one of addition, rather than one of subtraction. A better phrase would therefore be 'machine-learning augmentation', or one of similar meaning. The machine learning output depends critically on the training data. The training data here appears to be a fair approximation of a single crystal nanoparticle with an fcc structure. The models have both a fixed lattice parameter and thermal vibration B-factors, while one would expect these to be rather different for atoms on the surface of a real nanoparticle. However, I am satisfied that these are good enough for a starting model and the apparent observation of surface relaxation effects seems to bear this out. I am less sure of the removal of 0.5% of atoms 'to simulate atomic defects', which I calculate to be between 10 and 15 missing atoms in the nanoparticle sizes used here. The fraction of atoms located in error in Supplementary Fig. 2 is also around 0.5%. It therefore is essential to establish how much of this error is due to the atoms intentionally removed. If most of the error is in fact due to these atoms, it would seem that all the algorithm does is to fill the volume of the nanoparticle with a best-fit fcc lattice, which is not such a spectacular achievement. It would be useful to know how many atoms in the final reconstruction were not actually at fcc lattice coordinates. (Also on this note, why is RMS deviation used as a measure of the quality of fit between reconstruction and ground truth? Why not just mean amplitude of deviation?)

One would expect, given the limitations of the training data, that the algorithm would fail spectacularly on very small nanoparticles with amorphous structure, or larger nanoparticles that are commonly observed to be twinned. However, the inclusion of amorphous test data as shown in the 3rd column of Supplementary Fig. 2 seems to show that this is not the case, with atom location errors on average below 2%. I think it would be useful to give more information on these tests since it would show that the reconstruction is indeed based on an atomistic, rather than a lattice-based model. It would also be more realistic to simulate small nanoparticles of the order of 1nm diameter, since these would be amorphous in reality. Also, if the authors have any experimental data from larger twinned nanoparticles that have been successfully reconstructed, their inclusion would also make their claims much more convincing.

Not a lot of detail is given on the training process or how the number of 10,000 training data sets was arrived at. How many slices/images were used in the tomographic reconstructions? How do we know whether the training process has successfully converged, or has been over-fitted?

Inclusion of learning curves are necessary to establish the robustness of the process.

Supplementary Fig. 1(b) is clearly not the 3D Fourier intensity of the raw tomogram data, which would be restricted to lie on planes corresponding to the different orientations of the nanoparticle. There is significant intensity between these planes, which perhaps indicates that at least part of the success of the approach is due to the GENFIRE algorithm. Similarly there is intensity in the missing wedge, which shows that something has been done to replace the missing data. The origin of this effect/artefact should be explained.

Supplementary Fig. 3 is supposed to show the tracing error, but seems to compare structures produced from GENFIRE-reconstructed particles with GENFIRE+ML reconstructed particles. Since these are from simulations, shouldn't these plots show the difference between the two methods and ground truth? Also, it is not surprising that the errors for atoms on the surface of the particles are smaller than those in the bulk of the material? In fact, I find it difficult to believe that a reconstruction fidelity of 93% can be obtained from a sparse reconstruction covering only +/-65 degrees in the absence of ML and so this makes me wonder again how many different images were used. It is essential to give this information since it makes the difference between a small improvement when dealing with perfect simulated data and an algorithm that is actually useful for experimental data.

In summary, this is really nice work that could be published, but some more details are needed to establish that it is sound and robust.

Reviewer #2 (Remarks to the Author):

A new method for analyzing the surface atomic structures of nanomaterials presented by J. Lee et al. is important development for the emerging field of 3D TEM tomography of nanomaterials. Surface atomic structures are especially important for catalytic noble metal heterogeneous catalysts, but has been very challenging to deconvolute from the experimentally reconstructed maps. Since this manuscript addresses how we can approach to this important problem, it is worth being published. However, it still requires important validations of their method before publication.

1. Nanoparticles are exposed to the high vacuum during a long image acquisition in TEM tomography which can potentially transform positions of surface atoms. It is remarkable to obtain high resolution maps and to utilize neural network to identify atomic positions, but the results does not represent realistic surface structures due to this intrinsic limitation in microscopic method used in the study. Even if the transformation of atomics positions could be in very small scale such as a few to tens of pm, that is enough to change surface induced properties including adsorption, desorption, and coordination chemistry during catalytic reactions.

2. The authors used the simulated tomograms to train the DL filter and implemented it to denoise the experimental tomograms, as a way to confirm the results in the manuscript. To support applicability of the DL filter to experimental data, the quantitative comparison between the simulated STEM images and the experimental images is needed.

3. In supplementary figure 11, the strain factors at the surface still shows heterogeneity among the different models, which leaves a question regarding the unbiased decision for obtaining the most reliable strain map – in other words, what are factors that should be considered to choose the optimal DL filter for experimental tomograms? Following concern is that the simulated tomogram trained DL filter creates ambiguity in evaluating the optimal DL filter for experimental tomograms – more specifically, the tracing error and RMSD obtained from simulated data in Supplementary Figure 3 are not appropriate for validating the models for experimental data, and the tracing error and RMSD obtained from experimental data in Supplementary Figure 9 are also not valid because the difference between before and after the DL filtering does not tell us whether the model picks the right position of atoms.

4. It is uncertain that the weight sharing convolution layer based U-Net filter could work well with the tomograms with missing-wedge problem. Several tomograms in the figures show higher ratio of noises and artifacts along the wedge direction, which means that each region follows different noise distribution. However, the weight sharing convolution layer usually assumes that the nature of each patches is the same across the whole region. This could yield the difference in filtering at each different region, which poses a problem in the fidelity of filtered atom intensity profile. Therefore, to check whether the DL filter well resolves the elongation regardless of where the atoms are located, the supporting data in addition to the supplementary figure 6 is required.

5. In 'strain map calculation' section in Method, authors explain that '[100] direction showing more <100> like behavior, and [010] directions showing more <111> behavior', which is based the

difference in the number of atoms in (± 100) and (0 ± 10)-facets. However, the number of atoms in (100) is 35 and those in (010) is 34. Therefore, I think the authors' explanation is overstated. To support the anisotropic features of strain tensors, strain map calculation along $\langle 111 \rangle$ -direction is needed, by properly rotating 3D structure to place x-axis to eight different $\langle 111 \rangle$ -directions.

6. Fig 4 is unclear and make it difficult for readers from broad research areas to understand. For fig 4a, b, and c, overlapping with 3D structure and SiN support will be helpful. In addition, for fig 4d and e, it is hard to figure out an association of sliced maps and their locations with respect to the 3D structure and the position of SiN support. More detailed depiction in the Figure should be needed.

Reviewer #3 (Remarks to the Author):

Dear Editor,

Lee and coworkers describe a method for using deep learning to assist in the reconstruction of the 3D atomic structure of nanoparticles. While there have recently been numerous publications of deep-learning assisted reconstruction of tomography data, it has to my knowledge been limited to larger length-scales, typically in the context of medical images or composite materials. There have also recently been a number of publications using deep learning to assist in the interpretation of 2D STEM and TEM images. In my opinion, the combination presented in this manuscript is sufficiently new to warrant publication in Nature Communications, in particular as it provides a significant enhancement to the state-of-the-art in 3D atomic resolution tomography, where in particular the missing wedge problem otherwise limits the usability of the obtained data.

While my overall evaluation of the manuscript is positive, there are a number of issues that should be addressed before publication.

First and foremost is the question of possible bias introduced by the training set, and its influence on the final result. It appears that the authors have the data to address this, but it should be more explicit. When training the neural network on data where all atoms are in FCC lattice sites, the network will develop a bias for placing atoms in FCC sites. This is particularly troublesome as the main application (according to the introductory paragraph and the first paragraphs of the main text) is to look for non-trivial surface reconstructions and relaxations, as these may be crucial for the functionality of the nanoparticle. A bias from the network towards fcc structures would cause the reconstruction to be likely to miss non-trivial surface reconstructions in favor of interpreting the data as atoms on a lattice. Neural networks unfortunately tend to fill in data in areas where there is no data consistent with the training set.

The authors *do* address this, by also training networks on amorphous structures and showing that they are still able to analyse simulated data of fcc-based nanoparticles with almost as good results as when the network is trained on fcc-based particles. It must be assumed that the opposite is not the case: that the network trained on fcc-based particles will fail utterly at reconstructing the amorphous nanoparticles.

The point where I would like to see this addressed a bit more explicitly is in the analysis of the experimental data. Although it is not clear in the manuscript, it is my impression that the experimental data is analysed using a network trained on fcc nanoparticles. It is therefore not surprising that it is able to "clean up" the reconstruction of the atomic structure and assign the atoms to fcc sites. However, if the nanoparticle actually contains surface reconstructions that move atoms far from the fcc sites, the network would not be expected to recognize this. Furthermore, it is not clear to what extent the measured surface relaxations will be affected by the introduction or absence of systematic surface relaxations in the training data, if the training set only contains atoms randomly perturbed from the equilibrium fcc site but not any systematic surface relaxations, it is possible that the neural network will be biased by this and underestimate the surface relaxations.

A way to address this would be to use the network trained on amorphous data to analyse the experimental nanoparticle. As that network is still able to reconstruct simulated fcc-based nanoparticles, I assume that it will also be able to reconstruct the structure of the nanoparticle from the experiment (and if not, that is an important negative result that must be reported). It is likely that the reconstruction will look somewhat less nice than the current results, but it should be less biased towards "nice-looking" structures (but of course possibly bias in the opposite direction). I therefore strongly suggest that the authors repeat the analysis of the nanoparticle, including the measurements of surface relaxation, with an analysis based on the other network. The authors could even move the current figure 3 to the supplementary section, and explicitly state how many differences they find using the two networks, and also if the surface strain measurements are significantly affected.

While this is my main criticism against the manuscript, I would also like to mention a few other things that could be improved:

In line 118-129, and in the Methods section line 444-453, the authors seem to be doing an image simulation of the reconstructed model of the nanoparticle, and then reanalysing this with the network. The purpose of this is not entirely clear to me. Is this a normal "sanity check" when doing tomography? The purpose of doing this should be more clearly stated, as should the conclusion from this part of the work.

In lines 136-138 the out-of-plane displacements of the surface atoms for various facets are given, with an uncertainty that is between two and seven times the value. I hope and assume that this is the standard deviation of the positions of the individual atoms, not the uncertainty on the mean value. The latter should be reported as well.

In line 163-164 it is stated that "we demonstrated that the atomicity-based approach can reliably identify the surface atomic structure with a precision of 15 pm." Where does this value of 15 pm come from?

I am not confident that I can reconstruct the network architecture from Figure 1 and the description in the Methods section, although this is of course mitigated by the authors planning to publish the code. I am slightly confused by Figure 1. In the encoding (downsampling) path of the U-Net, we see alternations between convolutions and maxpool downsamplings, but the convolutions are labeled "downsampling" in the legend. In the decoding path, we see two kinds of convolutions, one labeled "upsampling". Are the convolutions in the encoding path also downsampling (with a stride?), or is it only the maxpool layers that downsample? And is the upsampling done by the convolutions, or by explicit upsampling layers? Finally, do each arrow represent a single convolution layer, or a stack of layers?

I am not completely sure how to read figure 4 and supplementary figure 11. Are the colored ellipsoids slices through the nanoparticle?

Finally, I have a few minor corrections:

In line 60, after mentioning the "atom-tracing method" please add a ref to the method section.

In line 78, I assume the RMSD is the RMSD of the atomic positions, but this should be stated explicitly.

Line 81: "... structure with different Gaussian width". Which Gaussian is this referring to?

Figure 2: Panel labels i and j are swapped in the caption.

Line 334: "the step 1 above". It is not clear exactly what this refers to, as there are no explicitly labelled steps.

Line 355: Why are no C3 and C5 aberrations included, in particular C3 could influence the images.

Or is it so small in the experimental setup that it is insignificant, in which case it might be worth stating it.

The "Facet analysis" is a bit hard to read. What is the meaning of the sentence "which results in [100] directions showing more $\langle 100 \rangle$ like behavior, and [010] directions showing more $\langle 111 \rangle$ like behavior." ?

Something is wrong in the legends in Supplementary figure 3 (a) and (c). In (a), shouldn't it be "Before DL" instead of "Before and After DL"? And in (c), should the network labels be the same as in (a), i.e. "FCC Bf5", "Amorphous" and "FCC Bf3.4"? Also, it is not clear if this is for the linear data or for the PRISM data.

Supplementary figure 9: Same comment as Suppl Fig 3.

Finally, I find it a bit surprising that a paper about atomic-resolution electron tomography does not reference some of the ground-breaking work from the University of Antwerp. It could for example be Van Aert et al, Nature 470, 364 (2011), www.nature.com/doi/10.1038/nature09741 or Goris et al., Nano Letters, 15, 6996 (2015) www.nature.com/doi/10.1038/nature09741
Disclaimer: I have no stakes in these publications.

First, we thank the referees for taking the time and effort to provide constructive comments, which have been very helpful in making improvements to the manuscript.

Please find below our response to the reviewers' comments on our manuscript "Single-atom level determination of 3-dimensional surface atomic structure via neural network-assisted atomic electron tomography". We have carefully addressed all the referees' criticisms and suggestions and trust that with the presented changes, the manuscript is now acceptable for publication in *Nature Communications*.

Response to reviewers' comments

Reviewer #1:

This article demonstrates a quite spectacular increase in the fidelity of three-dimensional reconstruction of fcc metallic nanoparticles using a machine learning approach in combination with relatively sparse aberration-corrected scanning transmission electron microscopy (STEM) tomograms. The improvement is primarily a result of the application of a three-dimensional unet machine-learning CNN algorithm, trained on PRISM-simulated data.

- 1. The reconstructed data is certainly impressive, but a great deal of information is lacking that would convince the reader that the results are reliable. I also find the description of the procedure used as a 'machine-learning filter' to be misleading and unsatisfactory. A filter, in the usual sense of the word, removes some component, separating it from another component that is of more use or interest. This applies to filtration of materials from liquids, removal of contaminants, improving signal to noise ratio or choosing a particular frequency, or energy, of some radiation or other signal. The process here is quite different; here, a machine learning algorithm fabricates some imagined data to fill gaps in a sparse data set. The fidelity of this process, when applied to test or experimental data, relies on the accumulated statistical accuracy of performing the same process on a large training data set. The process is thus one of addition, rather than one of subtraction. A better phrase would therefore be 'machine-learning augmentation', or one of similar meaning.*

Response:

We thank the referee for the very good suggestion. We agree that 'filter' may not be fully appropriate in the context of our work. As suggested by the referee, we changed the description of our deep-learning assisted procedure as 'deep-learning augmentation'.

- 2. The machine learning output depends critically on the training data. The training data here appears to be a fair approximation of a single crystal nanoparticle with an fcc structure. The models have both a fixed lattice parameter and thermal vibration B-factors, while one would expect these to be rather different for atoms on the surface of a real nanoparticle. However, I am satisfied that these are good enough for a starting model and the apparent observation of surface relaxation effects seems to bear this out. I am less sure of the removal of 0.5% of atoms 'to simulate atomic defects', which I calculate to be between 10 and 15 missing atoms in the nanoparticle sizes used here. The fraction of atoms located in error in Supplementary Fig. 2 is also around 0.5%. It therefore is essential to establish how much of this error is due to the atoms intentionally removed. If most of the error is in fact due to these atoms, it would seem that all the algorithm does is to fill the volume of the nanoparticle with a best-fit fcc lattice, which is not such a spectacular achievement. It would be useful to know how many atoms in the final reconstruction were not actually at fcc lattice coordinates. (Also on this note, why is RMS deviation used as a measure of the quality of fit between reconstruction and ground truth? Why not just mean amplitude of deviation?)*

Response:

We thank the referee for raising an important point regarding the generation of a training dataset. For usual metallic f.c.c. or b.c.c. crystals, the level of vacancy defects is very low at room temperatures^{1,2}. The training atomic models with 0.5% defect therefore included only a few missing atoms in the lattice, as the reviewer already pointed. To prove that the 0.5% error is not merely from the misidentified defects, we investigated the tomograms before and after the DL augmentation, especially the atomic layers which contain vacancy defects. As shown in Rev. Fig. 1, the raw reconstruction can already identify the defects, and the DL augmentation identifies the defect even more clearly. This shows that the internal vacancy defects can be successfully determined by tomographic reconstruction and the DL augmentation, not being the sole source of the error we report.

To further test the robustness of our approach, we also generated test datasets with vacancy defect levels of 5%, 10% and 20% separately. Models with defect levels of 5%, 10% and 20% can have the number of defects randomly chosen between 0 and 5%, 10%, and 20%, respectively. As shown in Rev. Fig. 2, in all three cases, tracing errors after the DL augmentation were dramatically reduced compared to the tracing errors before the DL augmentation. The tracing errors after the DL augmentation are less 1% regardless of the defect level. Overall, surface tracing errors and RMSDs of different defect level models (5%, 10%, 20%) are comparable to those of 0.5% defect level case we report in the manuscript. Therefore, we claim that the performance of our DL augmentation is independent of the existence of the defects, and the 0.5% error we report is the genuine intrinsic error of our approach, not from the 0.5% defect we included in the training model. We have included this discussion about the defect level dependence in lines 133-169 and 555-578 in our revised manuscript.

Our training and test atomic models based on the f.c.c. structures are actually not ideal f.c.c. models, since we included small spatial displacements to all atoms (RMSD: 22 pm). Tracing errors of our simulations are calculated based on how many traced atoms are not in the ground truth which is an f.c.c.-based structure with 22 pm RMSD displacements. For the case of experimental data, we assigned the measured atom positions to ideal f.c.c. lattices sites, which is conceptually different from the tracing error in the simulations. About 97% of traced atoms were assigned in ideal f.c.c. lattice sites in case of the experimental data, which is another evidence showing that the 0.5% error reported here is not an artifact from defect sites being filled in to become an ideal f.c.c. structure.

As the reviewer pointed out, the mean amplitude of deviation (MAD) is one of the good measures for overall atomic deviation. But RMSD of atomic positions is widely exploited as a measurement of an averaged atomic deviation, because in many cases the final atomic structures are fitted with ideal lattices to determine the atomic displacements and related strain¹⁻⁴. Since most of the minimization-based fitting algorithms are based on least-square figure of merits, they effectively minimize RMSD rather than MAD. We also calculated the MAD in addition to the RMSD from a simulated test dataset. The MAD and RMSD between the DL augmented traced atomic positions and the ground truths from the test dataset of 1,000 tomograms were obtained as 28.6 pm and 34.1 pm, respectively. Since RMSD is more sensitive to outliers compared to MAD, RMSD usually shows a larger value than MSD, as we observe here. Therefore, by using RMSD, we are providing a more conservative numbers for the measure of the atomic deviations.

Revision Figure 1 | Identified defects in the atomic layer slices of the simulated 3D tomographic reconstructions before and after the DL augmentation. a-c The atomic layers near the core of the nanoparticle. **d-f** The atomic layers near the surface of the nanoparticle. The layers are from ground truth (**a,d**), raw tomogram (**b,e**) and tomogram after applying the DL augmentation (**c,f**). Red circles highlight the locations of vacancy defects. Grayscale background represents the reconstructed intensity, and blue dots represent the positions of traced Pt atoms. Scale bar, 1 nm.

Revision Figure 2 | Tracing errors and RMSDs of atomic structures obtained from the simulations with different defect levels. The tracing errors (a,b) and RMSDs (c,d) were calculated by comparing the ground truths and traced atomic models from linear projection-based simulations before and after the DL augmentation. Each bar was obtained by averaging the results from the test sets of 100 tomograms (except for the 0.5% defect level case where we used the test dataset of 1,000 tomograms). The error bars represent the standard deviations. The DL augmentation neural network was trained by the training dataset based on f.c.c atomic models with defect level 0.5% and Bfactor 5.

3. *One would expect, given the limitations of the training data, that the algorithm would fail spectacularly on very small nanoparticles with amorphous structure, or larger nanoparticles that are commonly observed to be twinned. However, the inclusion of amorphous test data as shown in the 3rd column of Supplementary Fig. 2 seems to show that this is not the case, with atom location errors on average below 2%. I think it would be useful to give more information on these tests since it would shown that the reconstruction is indeed based on an atomistic, rather than a lattice-based model. It would also be more realistic to simulate small nanoparticles of the order of 1nm diameter, since these would be amorphous in reality. Also, if the authors have any experimental data from larger twinned nanoparticles that have been successfully reconstructed, their inclusion would also make their claims much more convincing.*

Response:

We thank the referee for pointing out an important point. We first tested the small nanoparticle case by generating a test dataset of 100 tomograms of amorphous atomic models about 1 nm in diameter. They were prepared following the process for generating the amorphous models (Methods). The only difference is that each volume ranges from 3,400 to 5,700 Å³ (approximately 200-300 atoms) in this case. Revision Figure 3 shows the results before and after the DL augmentation for the small nanoparticle case. The intensity of the 3D tomogram after the DL augmentation in Rev. Fig. 3e,f shows significant improvement compared to that before DL augmentation. The red circles in Rev. Fig. 3a-c show that the traced atoms after the DL augmentation (Rev. Fig. 3c) are much closer to the ground truth (Rev. Fig. 3a). We also calculated the tracing errors and RMSDs of the test dataset of 100 tomograms, and they clearly decrease after the DL augmentation, as expected. The order of tracing errors after the DL augmentation is approximately 1-2%, which means that the number of misidentified atoms is only 1-5.

Secondly, for the larger twinned nanoparticle case, we were unable to conduct further experimental tests due to the limited access to the aberration-corrected electron microscopes caused by the recent COVID-19 situation. Instead, we did run a simulational test for this case. An atomic model of decahedral Pt nanoparticle with twin boundaries⁵ was used for generating a test dataset of 100 tomograms by applying random cropping/rotation/translation and vacancy defect insertion, following the procedure of generating the f.c.c.-based atomic models (Methods). As shown in Rev. Fig. 5, the intensities in the atomic layers after the DL augmentation could be clearly identified as individual atoms. For quantitative comparison, the tracing errors and RMSDs of the test dataset of 100 tomograms were calculated. Revision Figure 6 shows that the tracing errors and RMSDs after the DL augmentation were significantly improved. Note that for this simulation of decahedral Pt nanoparticle, 27 tilt series images were used (instead of 21) and the tilt angles were chosen in the range between -75° and +75°, because resolving twin boundary structures required a higher-quality tomogram. We have included this discussion about testing small amorphous models and decahedral models with twin boundaries in lines 133-169 and 555-578 of our revised manuscript.

Revision Figure 3 | Traced atomic models and iso-surface renderings from tomography simulations of 1 nm diameter amorphous nanoparticle, before and after the DL augmentation. Traced atomic positions from the 3D volumes of ground truth (a), simulated 3D tomographic reconstructions before (b) and after the DL augmentation (c). The 3D iso-surface of ground truth (d), simulated 3D tomographic reconstructions before (e) and after the DL augmentation (f). The 3D iso-surfaces were plotted with 10% (d,f) and 30% (e) intensity threshold from the maximum intensity. The red circles highlight the location of misidentified atoms in the raw reconstruction, which can be clearly resolved after the DL augmentation. The DL augmentation neural network was trained by training dataset based on f.c.c atomic models with Bfactor 5. Scale bar, 1 nm.

Revision Figure 4 | Tracing errors and RMSDs of the atomic structures obtained from the simulated reconstructions of 1 nm diameter amorphous nanoparticles. The tracing errors (a,b) and RMSDs (c,d) were calculated by comparing the ground truths and traced atomic models from the linear projection-based tomographic simulations before and after the DL augmentation. Each bar was obtained by averaging the results from the test dataset of 100 tomograms. The DL augmentation neural network was trained by the training dataset based on f.c.c atomic models with Bfactor 5. The error bars represent the standard deviations.

Revision Figure 5 | Effect of the DL augmentation in the atomic layer slices of a simulated 3D tomogram of a decahedral nanoparticle with twin boundaries. a-f 2-Å thick slices perpendicular to [001] direction, obtained from the 3D tomogram of the nanoparticle near the center region (a-c) and near the surface (d-f). The slices are obtained from the ground truth (a,d), raw tomogram (b,e) and tomogram after applying the DL augmentation (c,f). Red circles highlight the locations of misidentified atoms in the raw reconstruction, which can be clearly resolved after the DL augmentation. Grayscale background represents the reconstructed intensity, and blue dots represent the positions of traced Pt atoms. The DL augmentation neural network was trained by the training dataset based on f.c.c atomic models with Bfactor 5. Scale bar, 1 nm.

Revision Figure 6 | Tracing errors and RMSDs of the atomic structures obtained from the simulated reconstructions of decahedral nanoparticles with twin boundaries. The tracing errors (a,b) and RMSDs (c,d) were calculated by comparing the ground truths and traced atomic models from linear projection-based tomographic simulations before and after the DL augmentation. Each bar was obtained by averaging the results from the test dataset of 100 tomograms. The DL augmentation neural network was trained by the training dataset based on f.c.c atomic models with Bfactor 5. The error bars represent the standard deviations.

4. *Not a lot of detail is given on the training process or how the number of 10,000 training data sets was arrived at.*

Response:

Thank you for raising a good point. We did monitor the loss function during the training and verified that the trainings were successfully converged. We tested multiple input training datasets, consisting of various numbers of 3D volumes, and the use of 10,000 training volumes was good enough to get the best performance of the DL augmentation. To support our claim, we show the loss function evolution curves obtained from the trainings with 5,000 and 10,000 training volumes. The evolutions of the validation losses from the DL networks trained by 5,000 and 10,000 tomograms show similar level of the loss function (Rev. Fig. 7). We mentioned the information about this training process in lines 613-617 in the revised manuscript, and also included the loss function curves in Supplementary Fig. 21.

Revision Figure 7 | Learning curves of the DL augmentation networks trained by different number of training datasets. The learning curves represent the validation losses as a function of training epochs. The blue and red lines show the validation losses during the trainings with 5,000 and 10,000 tomograms, respectively.

5. *How many slices/images were used in the tomographic reconstructions?*

Response:

The number of projections of both simulated and experimental data is 21. We clarified this in the method of our revised manuscript (line 511, Methods, 'Generation of input and target datasets for neural network training').

6. *How do we know whether the training process has successfully converged, or has been over-fitted? Inclusion of learning curves are necessary to establish the robustness of the process.*

Response:

This is a very good point, and we fully agree that we should have included the learning curves in the manuscript to show the convergence of the deep learning training process. Therefore, we show the learning curves for three different DL networks in Rev. Fig. 8. There is no overfitting in all three learning curves and all training processes were converged after approximately 50-70 epochs. We have included this discussion about the training process and learning curves in lines 618-621 in our revised manuscript and Supplementary Fig. 22.

Revision Figure 8 | Learning curves of the DL augmentation networks for three different training datasets. The learning curves during the training of the networks based on (a) the f.c.c. atomic models with Bfactor 5, (b) the amorphous atomic models with Bfactor 5, and (c) the f.c.c. atomic models with Bfactor 3.2. The blue and red solid lines represent the losses from training datasets and validation datasets, respectively.

7. *Supplementary Fig. 1(b) is clearly not the 3D Fourier intensity of the raw tomogram data, which would be restricted to lie on planes corresponding to the different orientations of the nanoparticle. There is significant intensity between these planes, which perhaps indicates that at least part of the success of the approach is due to the GENFIRE algorithm. Similarly there is intensity in the missing wedge, which shows that something has been done to replace the missing data. The origin of this effect/artefact should be explained.*

Response:

Thank you for raising a good point. As the reviewer mentioned, the intensity in the missing wedge of the raw tomogram was partially recovered by the GENFIRE algorithm⁶. The algorithm iterates between real and Fourier space with constraints, which discard unphysical information in real space (by applying known constraints such as positivity or support constraint) and enforce the measured tilt series information in Fourier space. The iterative approach enables to partially fill the data between the measured tilt angles and in the missing wedge (See the details in GENFIRE paper⁶).

8. *Supplementary Fig. 3 is supposed to show the tracing error, but seems to compare structures produced from GENFIRE-reconstructed particles with GENFIRE+ML reconstructed particles. Since these are from simulations, shouldn't these plots show the difference between the two methods and ground truth?*

Response:

This is a good point. As the referee mentioned, the ground truth does exist in simulations, and the comparisons between simulated results and ground truths are presented in the Supplementary Fig. 2. The purpose of the analysis in Supplementary Fig. 3 (the comparison between before and after the DL augmentation, which is now Supplementary Fig. 10 in our revised manuscript) was to provide a guideline or reference for the expected tracing error of an actual experimental case where we do not have the ground truth. As shown in Supplementary Fig. 9 (now Supplementary Fig. 17 in our revised manuscript), we did the same analysis for the experimental case and verified that the experimental results indeed show similar behavior expected from the simulations.

9. *Also, it is not surprising that the errors for atoms on the surface of the particles are smaller than those in the bulk of the material?*

Response:

This is another good point and we thank the referee for pointing this out. During our analysis, we found that there are ambiguities in defining “surface” atom tracing error. In many cases, it is not easy to say if certain atoms are surface atoms or not when we compare two different atomic structures. Therefore, to define the surface atom tracing error, we only considered the surface atoms in one atomic structure. For a surface atom in one atomic structure, if there is an atom in the other atomic structure at the corresponding position, the surface atom was counted as a correctly traced surface atom. (A detailed procedure of calculating the surface error is described in Methods section, ‘Calculation of tracing error’ part). This resulted in smaller numbers for the surface atoms errors compared to the bulk ones, but this definition of the surface atom error can unambiguously quantify the fidelity of the surface atomic structures obtained from different tomograms.

10. In fact, I find it difficult to believe that a reconstruction fidelity of 93% can be obtained from a sparse reconstruction covering only +/-65 degrees in the absence of ML and so this makes me wonder again how many different images were used. It is essential to give this information since it makes the difference between a small improvement when dealing with perfect simulated data and an algorithm that is actually useful for experimental data. In summary, this is really nice work that could be published, but some more details are needed to establish that it is sound and robust.

Response:

This is an important point. The reviewer’s concern is reasonable if we only consider the quality of the reconstructed 3D volumetric tomogram itself. However, our final goal is retrieving the atomic coordinates from the 3D tomograms. We do this by tracking the local maxima positions from the tomograms. Even though raw tomograms with missing data can be quite a bit different from ideal tomograms due to the missing wedge effect (irregularly shaped or elongated atoms), most of the local maxima positions are still preserved in the tomogram. This is how the approximately 20 pm positional uncertainty and more than 90% fidelity of atom identification can be achieved^{2,4}.

Reviewer #2:

new method for analyzing the surface atomic structures of nanomaterials presented by J. Lee et al. is important development for the emerging field of 3D TEM tomography of nanomaterials. Surface atomic structures are especially important for catalytic noble metal heterogeneous catalysts, but has been very challenging to deconvolute from the experimentally reconstructed maps. Since this manuscript addresses how we can approach to this important problem, it is worth being published. However, it still requires important validations of their method before publication.

- 1. Nanoparticles are exposed to the high vacuum during a long image acquisition in TEM tomography which can potentially transform positions of surface atoms. It is remarkable to obtain high resolution maps and to utilize neural network to identify atomic positions, but the results does not represent realistic surface structures due to this intrinsic limitation in microscopic method used in the study. Even if the transformation of atomics positions could be in very small scale such as a few to tens of pm, that is enough to change surface induced properties including adsorption, desorption, and coordination chemistry during catalytic reactions.*

Response:

We thank the referee for pointing out an important issue. As the reviewer mentioned, the surface atomic structure can change at the high-vacuum environment of the TEM column, and also due to the electron bombardment during the measurement. Actually, the surface of metallic nanoparticles continuously changes at room temperature even without any external stimuli due to the surface diffusion at 10^{-4} - 10^{-7} s timescale⁷. The best way to truly reveal the static 3D surface atomic structure will be to run an ultrafast tomography measurement utilizing short-pulsed electron beams, but this is very challenging at current electron optics and detector technology. The surface structure we report may not be the fully static atomic structure at 10^{-4} - 10^{-7} s timescale⁷, but it can still provide the time average of the surface structure. Since hundreds of surface atoms at 14 facets of a Pt nanoparticle is being reported here, the average structure can still provide important information which can be useful for studying the surface catalytic activity. We would like to emphasize again that this is the only technique which can provide surface atomic structure at tens of picometer scale accuracy (although time averaged), as far as we know.

- 2. The authors used the simulated tomograms to train the DL filter and implemented it to denoise the experimental tomograms, as a way to confirm the results in the manuscript. To support applicability of the DL filter to experimental data, the quantitative comparison between the simulated STEM images and the experimental images is needed.*

Response:

Thank you for the good suggestion. In our manuscript, the crystallographic R-factor⁸ was used to quantitatively compare the experimental STEM images and simulated STEM images which were used for the DL network training. The calculated R-factors were determined to be 0.173 (DL, f.c.c. & Bfactor 5), 0.182 (DL, amorphous & Bfactor 5), 0.176 (DL, f.c.c. & Bfactor 3.2) and 0.192 (not DL) as described in lines 229-231 and 705-734 and Supplementary Fig. 23 of our revised manuscript. Note that R-factor smaller than 20% is considered as an indicator of a good agreement between the experiment and simulation⁹⁻¹¹.

- 3. In supplementary figure 11, the strain factors at the surface still shows heterogeneity among the different models, which leaves a question regarding the unbiased decision for obtaining the most reliable strain map – in other words, what are factors that should be considered to choose the*

optimal DL filter for experimental tomograms? Following concern is that the simulated tomogram trained DL filter creates ambiguity in evaluating the optimal DL filter for experimental tomograms – more specifically, the tracing error and RMSD obtained from simulated data in Supplementary Figure 3 are not appropriate for validating the models for experimental data, and the tracing error and RMSD obtained from experimental data in Supplementary Figure 9 are also not valid because the difference between before and after the DL filtering does not tell us whether the model picks the right position of atoms.

Response:

This is a very important point. As the reviewer pointed, our final surface structures of the Pt nanoparticle can be slightly different depending on the basis of training datasets for the DL neural networks (Supplementary Fig. 7,9,11 of the original manuscript and Supplementary Fig. 14,17,19 of our revised manuscript). We are not claiming that our final reported atomic structure is a definite error-free structure. The final atomic structure does suffer from uncertainties which can be caused by nonlinearity of STEM imaging, experimental noises, missing wedge effect, intrinsic errors in our DL augmentation approach, etc. This is why we have run several tests including multislice-based image simulations and applying neural networks trained by different basis structures. As the reviewer pointed, there is the heterogeneity among the different models (which is within the level of the structural uncertainty of 15 pm we report). However, from the simulations, we also find that if the basis atomic structure for DL network training is more similar to the test dataset, the result is always better than using the DL networks trained by different basis structure. For correcting the tomograms based on f.c.c. structure, the DL network trained with f.c.c.-based structure worked the best (Supplementary Fig. 2). In our additional test using amorphous based atomic models and f.c.c.-based atomic models with different spatial displacement in core atoms and surface atoms (Rev. Fig. 9,10), the same trend was observed. Further, the atomic structure obtained from the DL augmentation trained by f.c.c.-based atomic structure with Bfactor 5 shows the lowest R-factor, which indicates the best consistency with the experimental STEM images. Therefore, we selected the result from the DL network trained by the f.c.c.-based atomic models with Bfactor 5 as our final reported atomic structure.

Revision Figure 9 | Tracing errors and RMSDs of the tomography simulation results from amorphous atomic structures. The amorphous atomic structures for this test were generated using the same method for obtaining the amorphous training dataset (Methods). The tracing errors (a,b) and RMSDs (c,d) were calculated by comparing the ground truths and traced atomic models from linear projection-based tomographic simulations before and after the DL augmentation. Two different DL networks, one trained by the f.c.c.-based atomic models (Bfactor 5) and another trained by the amorphous models, were used for the

test. Each bar was obtained by averaging the results from the test datasets of 100 tomograms. The error bars represent the standard deviations.

Revision Figure 10 | Tracing errors and RMSDs of the tomography simulations of f.c.c. based atomic structures with spatial displacement of 22 pm for core atoms and 50 pm for surface atoms. The tracing errors (a,b) and RMSDs (c,d) were calculated by comparing the ground truths and traced atomic models from linear projection-based tomographic simulations before and after the DL augmentation. Two different DL networks, one trained by f.c.c. atomic models (Bfactor 5) and another trained by amorphous models, were used for the test. Each bar was obtained by averaging the results from the test datasets of 100 tomograms. The error bars represent the standard deviations.

4. *It is uncertain that the weight sharing convolution layer based U-Net filter could work well with the tomograms with missing-wedge problem. Several tomograms in the figures show higher ratio of noises and artifacts along the wedge direction, which means that each region follows different noise distribution. However, the weight sharing convolution layer usually assumes that the nature of each patches is the same across the whole region. This could yield the difference in filtering at each different region, which poses a problem in the fidelity of filtered atom intensity profile. Therefore, to check whether the DL filter well resolves the elongation regardless of where the atoms are located, the supporting data in addition to the supplementary figure 6 is required.*

Response:

This is an important point. As the reviewer pointed, the same convolution kernel is being applied to the entire volume during the DL augmentation. This method can be justified in our case because we have the prior knowledge about our system: the *atomicity*. The essential information which our tomographic reconstruction (complicated 3D volume data) contains is the 3D coordinates of thousands of atoms in the nanoparticle. This means that our full 3D volume, which consists of millions of voxels, is redundant, and the essential information can be sparsely represented. Our DL architecture has two parts: encoder and decoder. The role of the encoder is to extract meaningful information related to the 3D atomic positions of the nanoparticle. While propagating through the well-trained neural network layers via max-pooling and non-linear activation function (Leaky ReLU), redundant information and undesired artifacts are being successfully filtered to only preserve the essential atomic structural information, and our decoder can successfully generate the noise-free atomic profiles at different regions using the sparse atomic structural information. Moreover, at the deepest u-net layer (the part between encoder and decoder), the size of feature map (9x9x9 voxels) becomes comparable to the convolution kernel size (3x3x3 voxels), and at this

stage, applying properly trained convolution kernel will be able to differently affect the missing wedge and non-missing regions.

We checked the elongation effects at different atomic positions. As shown in Rev. Fig. 11, we plotted slices of 3D atomic volume profiles for three different atoms; one located near the surface along the direction perpendicular to the missing wedge direction, one located near the surface along the missing wedge direction, and one located near the core of the nanoparticle, respectively. The three atoms show different elongation profiles before the DL augmentation but applying the DL augmentation can successfully retrieve the expected isotropic 3D Gaussian shape of the intensity profile. We have added this figure as Supplementary Fig. 16, and discussed this in lines 225-226 in our revised manuscript.

Revision Figure 11 | Representative 3D atomic profiles obtained from the experimental tomogram before and after the DL augmentation. **a** The atomic structure of the Pt nanoparticle after the DL augmentation. **b-c** Intensity profile (central slices) of an atom near the surface along the direction perpendicular to the missing wedge direction [the purple atom in (a)] before (b) and after (c) the DL augmentation. **d-e** Intensity profile (central slices) of an atom near the surface along the missing wedge direction [the green atom in (a)] before (d) and after (e) the DL augmentation. **f-g** Intensity profile (central slices) of an atom near the core of nanoparticle [the red atom in (a)] before (f) and after (g) the DL augmentation. The pixel size is 0.357 Å.

5. In 'strain map calculation' section in Method, authors explain that '[100] direction showing more <100> like behavior, and [010] directions showing more <111> behavior', which is based the difference in the number of atoms in (± 100) and (0 ± 10)-facets. However, the number of atoms in (100) is 35 and those in (010) is 34. Therefore, I think the authors' explanation is overstated. To support the anisotropic features of strain tensors, strain map calculation along <111>-direction is needed, by properly rotating 3D structure to place x-axis to eight different <111>-directions.

Response:

We thank the reviewer for the important suggestion. As the reviewer recommended, we plotted the strain maps along the $\langle 111 \rangle$ directions in Rev. Fig. 12. As shown Rev. Fig. 12, the xx-strains along both $[111]$ and $[1\bar{1}\bar{1}]$ directions represent tensile strain, but the x-strains along both $[11\bar{1}]$ and $[1\bar{1}\bar{1}]$ directions close to the SiN membrane represent a gradual change along the z-direction in the given axis convention. The exotic strain is caused by the substrate effect as we reported. Tensile strains are dominant in $\langle 111 \rangle$ facets except for the facets close to the substrate. Therefore, we feel more confident that the origin of overall tensile strain along the y-direction in $\langle 100 \rangle$ axis convention comes from the $\langle 111 \rangle$ facets. We have included the discussion in our revised manuscript in lines 301-307 and Supplementary Fig. 20.

Revision Figure 12 | 3D strain maps calculated from $\langle 111 \rangle$ axis conventions. The strain tensors calculated for four different axes conventions with respect to the f.c.c. crystallographic axes. The x-directions are (a) $[111]$, (b) $[1\bar{1}\bar{1}]$, (c) $[11\bar{1}]$, (d) $[1\bar{1}\bar{1}]$ directions, respectively. Clear tensile strain is observed along the $[111]$ and $[1\bar{1}\bar{1}]$ directions (ϵ_{xx} maps of (a) and (b)) which are at the opposite side of the particle-substrate interface. On the other hand, along the $[11\bar{1}]$ and $[1\bar{1}\bar{1}]$ directions (which are toward the substrate), compressive strains can be observed for the ϵ_{xx} maps. All strain maps were calculated from the atomic structure obtained from the experimental data after the DL augmentation trained by f.c.c.-based atomic model with Bfactor 5. Scale bar, 2 nm.

- Fig 4 is unclear and make it difficult for readers from broad research areas to understand. For fig 4a, b, and c, overlapping with 3D structure and SiN support will be helpful. In addition, for fig 4d and e, it is hard to figure out an association of sliced maps and their locations with respect to the

3D structure and the position of SiN support. More detailed depiction in the Figure should be needed.

Response:

We thank the reviewer for the constructive comments. Following the referee's suggestions, we revised the Fig. 4 and included the position of SiN support membrane with respect to the nanoparticle, and the slices of the whole atomic structure to help the readers to identify the corresponding atomic layer for each row of the strain map diagram.

Reviewer #3:

Lee and coworkers describe a method for using deep learning to assist in the reconstruction of the 3D atomic structure of nanoparticles. While there have recently been numerous publications of deep-learning assisted reconstruction of tomography data, it has to my knowledge been limited to larger length-scales, typically in the context of medical images or composite materials. There have also recently been a number of publications using deep learning to assist in the interpretation of 2D STEM and TEM images. In my opinion, the combination presented in this manuscript is sufficiently new to warrant publication in Nature Communications, in particular as it provides a significant enhancement to the state-of-the-art in 3D atomic resolution tomography, where in particular the missing wedge problem otherwise limits the usability of the obtained data. While my overall evaluation of the manuscript is positive, there are a number of issues that should be addressed before publication.

- 1. First and foremost is the question of possible bias introduced by the training set, and its influence on the final result. It appears that the authors have the data to address this, but it should be more explicit. When training the neural network on data where all atoms are in FCC lattice sites, the network will develop a bias for placing atoms in FCC sites. This is particularly troublesome as the main application (according to the introductory paragraph and the first paragraphs of the main text) is to look for non-trivial surface reconstructions and relaxations, as these may be crucial for the functionality of the nanoparticle. A bias from the network towards fcc structures would cause the reconstruction to be likely to miss non-trivial surface reconstructions in favor of interpreting the data as atoms on a lattice. Neural networks unfortunately tend to fill in data in areas where there is no data consistent with the training set.*

*The authors *do* address this, by also training networks on amorphous structures and showing that they are still able to analyse simulated data of fcc-based nanoparticles with almost as good results as when the network is trained on fcc-based particles. It must be assumed that the opposite is not the case: that the network trained on fcc-based particles will fail utterly at reconstructing the amorphous nanoparticles.*

The point where I would like to see this addressed a bit more explicitly is in the analysis of the experimental data. Although it is not clear in the manuscript, it is my impression that the experimental data is analysed using a network trained on fcc nanoparticles. It is therefore not surprising that it is able to "clean up" the reconstruction of the atomic structure and assign the atoms to fcc sites. However, if the nanoparticle actually contains surface reconstructions that move atoms far from the fcc sites, the network would not be expected to recognize this. Furthermore, it is not clear to what extent the measured surface relaxations will be affected by the introduction or absence of systematic surface relaxations in the training data, if the training set only contains atoms randomly perturbed from the equilibrium fcc site but not any systematic surface relaxations, it is possible that the neural network will be biased by this and underestimate the surface relaxations. A way to address this would be to use the network trained on amorphous data to analyse the experimental nanoparticle. As that network is still able to reconstruct simulated fcc-based nanoparticles, I assume that it will also be able to reconstruct the structure of the nanoparticle from the experiment (and if not, that is an important negative result that must be reported). It is likely that the reconstruction will look somewhat less nice than the current results, but it should be less biased towards "nice-looking" structures (but of course possibly bias in the opposite direction). I therefore strongly suggest that the authors repeat the analysis of the nanoparticle, including the measurements of surface relaxation, with an analysis based on the other network. The authors could even move the current figure 3 to the supplementary section, and explicitly state how many differences they find using the two networks, and also if the surface strain measurements are significantly affected.

Response:

This is a good suggestion. First, as the reviewer pointed out, we have emphasized the use of the DL augmentation trained on f.c.c.-based atomic structure in the experimental section in our revised manuscript.

Our answer to the possible bias toward perfect f.c.c. structure consists of two parts: simulations and experiments. In simulations, Supplementary Fig. 2,3 (Supplementary Fig. 2,10 in our revised manuscript) and Rev. Fig. 9,10 shows the robustness regardless of the basis of the training datasets (different Bfactors, different spatial displacements, or even amorphous based structure), as the reviewer already mentioned. In the case of the experiment, we have also tried applying the DL augmentation trained by amorphous models based simulated tomograms as the reviewer suggested (Supplementary Fig. 5,7 in our original manuscript, Supplementary Fig. 12,14 in our revised manuscript). Supplementary Fig. 6 (Supplementary Fig. 13 in our revised manuscript) clearly shows that the reconstruction artifacts can be successfully removed even by the use of the amorphous based DL augmentation. Furthermore, as illustrated in Supplementary Fig. 9 (Supplementary Fig. 17 in our revised manuscript), there are minor differences between the atomic structures obtained from differently trained DL augmentations, but the differences are smaller than those between the raw (not DL augmented) and DL augmented atomic structures. This clearly shows that even for the experimental data, the DL augmentation approach produces consistent results regardless of the basis of the training datasets (it can be amorphous based, or f.c.c.-based, or with the different Bfactor).

2. *While this is my main criticism against the manuscript, I would also like to mention a few other things that could be improved:*

In line 118-129, and in the Methods section line 444-453, the authors seem to be doing an image simulation of the reconstructed model of the nanoparticle, and then reanalysing this with the network. The purpose of this is not entirely clear to me. Is this a normal "sanity check" when doing tomography? The purpose of doing this should be more clearly stated, as should the conclusion from this part of the work.

Response:

The purpose of the analysis described in lines 118-129 (Methods section lines 444-453) is to estimate the precision of the atomic coordinates obtained from our approach, and this method is widely being used in atomic electron tomography research^{1,2,12}. We have included this information in lines 754-755 in our revised manuscript.

3. *In lines 136-138 the out-of-plane displacements of the surface atoms for various facets are given, with an uncertainty that is between two and seven times the value. I hope and assume that this is the standard deviation of the positions of the individual atoms, not the uncertainty on the mean value. The latter should be reported as well.*

Response:

Thank you for checking the detail. The uncertainty means the standard deviation of the 'displacements' of each atom. The values indeed seem to be quite large at first glance. However, the surface atoms typically show large deviations from their expected lattice positions. As can be seen Fig. 3,4, many surface atoms are largely displaced along many different directions from the perfect f.c.c. facet atomic sites. Therefore, the standard deviations of the surface atomic displacements are expected to be very large. Furthermore, as the reviewer mentioned, the real surface structure will be more like an amorphous structure rather than perfectly crystalline, and our

large standard deviation actually agrees with the expectation as well. We have included this explanation of the uncertainty in lines 811-812 in our revised manuscript.

4. In line 163-164 it is stated that "we demonstrated that the atomicity-based approach can reliably identify the surface atomic structure with a precision of 15 pm." Where does this value of 15 pm come from?

Response:

We performed the precision analysis as mentioned in the response to the 2nd comment of the 3rd reviewer, and the 15 pm is the estimated precision of the atomic structure obtained from the experimental tomogram after the DL augmentation. The procedure is described in the Methods section, 'Precision analysis for experimental data by PRISM STEM simulation' (lines 753-765 in our revised manuscript).

5. *I am not confident that I can reconstruct the network architecture from Figure 1 and the description in the Methods section, although this is of course mitigated by the authors planning to publish the code. I am slightly confused by Figure 1. In the encoding (downsampling) path of the U-Net, we see alternations between convolutions and maxpool downsamplings, but the convolutions are labeled "downsampling" in the legend. In the decoding path, we see two kinds of convolutions, one labeled "upsampling". Are the convolutions in the encoding path also down-sampling (with a stride?), or is it only the maxpool layers that downsample? And is the upsampling done by the convolutions, or by explicit upsampling layers? Finally, do each arrow represent a single convolution layer, or a stack of layers?*

Response:

We thank the reviewer for pointing out the unclear points. The reviewer is correct, the convolution layer (down-sampling) propagates with stride 2, so it reduces the volume size. Also, the convolution layer (up-sampling) consists of transpose convolution with stride 2. We have included more detailed discussion in lines 599-609 in our revised manuscript.

6. *I am not completely sure how to read figure 4 and supplementary figure 11. Are the colored ellipsoids slices through the nanoparticle?*

Response:

We thank the reviewer again for pointing out the unclear part of the figures. As the reviewer pointed out, it can indeed be confusing. The slices in Fig. 4 denote atomic layers along the [001] direction. We have modified the Fig. 4 to make this clear.

7. *Finally, I have a few minor corrections:*

Response:

We thank the reviewer for very detailed corrections.

In line 60, after mentioning the "atom-tracing method" please add a ref to the method section.

Response:

We have included the reference in our revised manuscript.

In line 78, I assume the RMSD is the RMSD of the atomic positions, but this should be stated explicitly.

Response:

We have modified the terminology clearly in our revised manuscript lines 124-126, following the reviewer's suggestion.

Line 81: "... structure with different Gaussian width". Which Gaussian is this referring to?

Response:

'Gaussian width' represents electron beam size (we assumed that the shape of the electron beam profile is 3D Gaussian distribution, as described in the original manuscript in lines 321-322). The Bfactor is the parameter controlling the 'Gaussian width'. The beam size depends on the value of Bfactor. We have added this explanation clearly in our manuscript lines 128-129 and 504-506.

Figure 2: Panel labels i and j are swapped in the caption.

Response:

We have corrected the typo in Fig. 3 caption in our revised manuscript.

Line 334: "the step 1 above". It is not clear exactly what this refers to, as there are no explicitly labelled steps.

Response:

We have modified the statement clearly in our revised manuscript.

Line 355: Why are no C3 and C5 aberrations included, in particular C3 could influence the images. Or is it so small in the experimental setup that it is insignificant, in which case it might be worth stating it.

Response:

As the reviewer pointed out, we fully agree that simulation including the C3 and C5 aberrations is a more correct way to mimic the true experimental images. The C3 and C5 aberrations obtained from the double Cs-corrected electron microscope which we used are -775 nm and 378 μm , respectively. We performed multislice-based PRISM calculations including these aberrations for the test dataset of 1,000 tomograms again. The result of the simulations does not show big difference compared to the previous result without the aberrations. Also, we re-did the PRISM-based precision analysis including the aberrations. To include the new results, we have modified the Fig. 2 and Supplementary Fig. 1,2, and lines 589-596 and 753-765 in the revised manuscript.

The "Facet analysis" is a bit hard to read. What is the meaning of the sentence "which results in [100] directions showing more <100> like behavior, and [010] directions showing more <111> like behavior." ?

Response:

The compressive and tensile strains are dominant in <100> and <111> facets, respectively. '<100> like behavior' and '<111> like behavior' denote compressive and tensile strain, respectively. We have modified this statement in our revised manuscript in lines 301-307.

Something is wrong in the legends in Supplementary figure 3 (a) and (c). In (a), shouldn't it be "Before DL" instead of "Before and After DL"? And in (c), should the network labels be the same

as in (a), i.e. "FCC Bf5", "Amorphous" and "FCC Bf3.4"? Also, it is not clear if this is for the linear data or for the PRISM data. Supplementary figure 9: Same comment as Suppl Fig 3.

Response:

Our purpose of Supplementary Fig. 3 is to compare atomic structures before and after the DL augmentations or after different DL augmentations. We already showed the results in Supplementary Fig. 2, as the reviewer mentioned. In Supplementary Fig. 3, linear projection simulations were exploited for all analyses. We have emphasized this point in Supplementary Fig. 3 caption (Supplementary Fig. 10 caption of our revised manuscript).

8. *Finally, I find it a bit surprising that a paper about atomic-resolution electron tomography does not reference some of the ground-breaking work from the University of Antwerp. It could for example be Van Aert et al, Nature 470, 364 (2011). www.nature.com/doi/10.1038/nature09741 or Goris et al., Nano Letters, 15, 6996 (2015) www.nature.com/doi/10.1038/nature09741 Disclaimer: I have no stakes in these publications.*

Response:

Thank you for your detailed feedback. Of course, we are aware of the important contribution to the field by the group at the University of Antwerp, and we did cite the Van Aert et al., *Nature* (2011) paper and Goris et al., *Nano Lett.* (2013) paper as references 5 and 6 in our original manuscript. As suggested by the reviewer, we additionally cited the "Goris et al., *Nano Lett.* 15, 6996 (2015)" paper as reference 10 in our revised manuscript.

References

1. Xu, R. *et al.* Three-dimensional coordinates of individual atoms in materials revealed by electron tomography. *Nature Materials* **14**, 1099–1103 (2015).
2. Yang, Y. *et al.* Deciphering chemical order/disorder and material properties at the single-atom level. *Nature* **542**, 75–79 (2017).
3. Coutsias, E. A., Seok, C. & Dill, K. A. Using quaternions to calculate RMSD. *Journal of Computational Chemistry* **25**, 1849–1857 (2004).
4. Zhou, J. *et al.* Observing crystal nucleation in four dimensions using atomic electron tomography. *Nature* **570**, 500–503 (2019).
5. Chen, C.-C. *et al.* Three-dimensional imaging of dislocations in a nanoparticle at atomic resolution. *Nature* **496**, 74–77 (2013).
6. Pryor, A. *et al.* GENFIRE: A generalized Fourier iterative reconstruction algorithm for high-resolution 3D imaging. *Scientific Reports* **7**, 1–12 (2017).
7. Schneider, S., Surrey, A., Pohl, D., Schultz, L. & Rellinghaus, B. Atomic surface diffusion on Pt nanoparticles quantified by high-resolution transmission electron microscopy. *Micron* **63**, 52–56 (2014).
8. Blundell, T. L. & Johnson, L. N. *Protein Crystallography*. (Elsevier Science, 1976).
9. Joosten, R. P., Chinae, G., Kleywegt, G. J. & Vriend, G. Protein Three-Dimensional Structure Validation. in *Comprehensive Medicinal Chemistry II* (eds. Taylor, J. B. & Triggle, D. J.) 507–530 (Elsevier, 2007). doi:10.1016/B0-08-045044-X/00096-1.
10. Rondeau, J.-M. & Schreuder, H. Protein Crystallography and Drug Discovery. in *The Practice of Medicinal Chemistry (Fourth Edition)* (eds. Wermuth, C. G., Aldous, D., Raboisson, P. & Rognan, D.) 511–537 (Academic Press, 2015). doi:10.1016/B978-0-12-417205-0.00022-5.
11. Krenkel, U. & Imberty, A. Crystallography and Lectin Structure Database. in *Lectins* (ed. Nilsson, C. L.) 15–50 (Elsevier Science B.V., 2007). doi:10.1016/B978-044453077-6/50003-X.
12. Tian, X. *et al.* Correlating the three-dimensional atomic defects and electronic properties of two-dimensional transition metal dichalcogenides. *Nature Materials* **19**, 867–873 (2020).

REVIEWERS' COMMENTS

Reviewer #1 (Remarks to the Author):

The changes to the article are a significant improvement over the initial submission and I am pleased to see that they have responded to all of my comments, and those of the other referees, in a constructive way. The inclusion of amorphous and twinned structures is really convincing and shows that the reconstruction is generally applicable to different structures. I hope to see it being used to tackle real experimental problems in the future.

Reviewer #2 (Remarks to the Author):

All concerns that I Asked are addressed well. Now, I support a publication of this manuscript in Nature Communications.

Reviewer #3 (Remarks to the Author):

As I wrote in the original referee report, it is my opinion that this work should be accepted for publication in Nature Communications. The work is a very timely combination of state-of-the-art electron tomography with machine learning, and a good example of a situation where machine learning can really make a difference for the field.

I raised a number of concerns in my original report, but it is my opinion that the authors have addressed these issues very thoroughly and convincingly. It is also my impression that they have addressed the concerns raised by the other reviewers.

The manuscript is now very readable, and it should be possible for people in the field of atomic-scale electron tomography to directly apply the methods presented here, in particular as the authors will make their data and source code available.

In conclusion, I recommend that the revised manuscript is accepted for publication in its present form.

Please find below our response to the reviewers' comments on our manuscript "Single-atom level determination of 3-dimensional surface atomic structure via neural network-assisted atomic electron tomography". We trust that the manuscript is now acceptable for publication in *Nature Communications*.

Response to reviewers' comments

Reviewer #1:

The changes to the article are a significant improvement over the initial submission and I am pleased to see that they have responded to all of my comments, and those of the other referees, in a constructive way. The inclusion of amorphous and twinned structures is really convincing and shows that the reconstruction is generally applicable to different structures. I hope to see it being used to tackle real experimental problems in the future.

Response:

We thank the reviewer for taking the time and effort to provide constructive comments, which have been very helpful in making improvements to the manuscript.

Reviewer #2:

All concerns that I Asked are addressed well. Now, I support a publication of this manuscript in Nature Communications.

Response:

We thank the reviewer for taking the time and effort to provide constructive comments, which have been very helpful in making improvements to the manuscript.

Reviewer #3:

As I wrote in the original referee report, it is my opinion that this work should be accepted for publication in Nature Communications. The work is a very timely combination of state-of-the-art electron tomography with machine learning, and a good example of a situation where machine learning can really make a difference for the field.

I raised a number of concerns in my original report, but it is my opinion that the authors have addressed these issues very thoroughly and convincingly. It is also my impression that they have addressed the concerns raised by the other reviewers.

The manuscript is now very readable, and it should be possible for people in the field of atomic-scale electron tomography to directly apply the methods presented here, in particular as the authors will make their data and source code available.

In conclusion, I recommend that the revised manuscript is accepted for publication in its present form.

Response:

We thank the reviewer for taking the time and effort to provide constructive comments, which have been very helpful in making improvements to the manuscript.